

# Is there an aerosol signature of cloud processing?

Barbara Ervens[1,2], Armin Sorooshian[3,4], Abdulmonam M. Aldhaif[3], Taylor Shingler[5,6], Ewan Crosbie[5,6], Luke Ziemba[6], Pedro Campuzano-Jost[2,7], Jose L. Jimenez[2,7], Armin Wisthaler[8,9]

[1] NOAA/ESRL/Chemical Sciences Division, Boulder, CO, USA
[2] CIRES, University of Colorado, Boulder, CO, USA
[3] Department of Chemical and Environmental Engineering, University of Arizona, Tucson, AZ, USA
[4] Department of Hydrology and Atmospheric Sciences, University of Arizona, Tucson, AZ, USA
[5] Science Systems and Applications, Inc., Hampton, VA, USA
[6] NASA Langley Research Center, Hampton, VA, USA
[7] Department of Chemistry, University of Colorado, Boulder, Colorado, USA
[8] Department of Chemistry, University of Oslo, Oslo, Norway
[9] Institute for Ion Physics and Applied Physics, University of Innsbruck, Innsbruck, Austria

*Correspondence to*: barbara.ervens@noaa.gov and armin@email.arizona.edu

**Abstract**

The formation of sulfate and secondary organic aerosol mass in the aqueous phase (aqSOA) of cloud and fog droplets can significantly contribute to ambient aerosol mass. While tracer compounds give evidence that aqueous phase processing occurred, they do not reveal the extent to which particle properties have been modified in terms of mass, chemical composition, hygroscopicity and oxidation state.

We analyse data from several field experiments and model studies for six air mass types (urban, biogenic, marine, wild fire biomass burning, agricultural biomass burning and background air). We focus on the trends of changes in mass, hygroscopicity parameter $\kappa$, and oxygen-to-carbon (O/C) ratio due to cloud processing. We find that the modification of these parameters upon cloud-processing is most evident in urban, marine and biogenic air masses, i.e. air masses that are more polluted than very clean air (background air) but cleaner than heavily polluted plumes as encountered during biomass burning. Based on these trends, we suggest that the mass ratio ($R_{tot}$) of the potential aerosol sulfate and aqSOA mass to the initial aerosol mass can be used to predict whether cloud processing will be detectable. Scenarios where this ratio exceeds $R_{tot} \sim 2$ are the most likely ones where clouds can significantly change aerosol parameters. Comparison to $R_{tot}$ values as calculated for ambient data at different locations confirm the applicability of the concept to predict a cloud-processing signature in selected air masses.

## 1 Introduction

Clouds and, in particular, aerosol-cloud interactions represent one of the largest uncertainties in our current understanding of radiative forcing (Stocker et al., 2013). Thus, representing cloud chemistry in models is challenging, as the prediction of aerosol mass production in clouds is inherently impacted by the uncertainties in the description of cloud properties (e.g., liquid water content (LWC), drop size distribution, cloud lifetime, geographical location, altitude and cloud density), in addition to uncertainties in the chemical mechanisms and precursors (Ervens, 2015). Airborne chemical measurements in clouds can be used to study cloud processing, but such measurements are



relatively sparse and usually only represent snapshots of a few seconds of an aircraft transect. Several studies have been performed on mountain tops where hill capped clouds cover the summit for extended period of times (Choularton et al., 1997; Herrmann et al., 2005; Li et al., 2017). While in "hill cloud" experiments more continuous data sets can

be collected, they are limited in their geographical coverage and their interpretation is complicated by variable advection of various sources and airmasses. Many studies show enhanced concentrations of sulfate, oxalate and related organics in cloud-processed air as compared to cloud-free air (Crahan et al., 2004; Sorooshian et al., 2006a; Sorooshian et al., 2007a; Wonaschuetz et al., 2012). It has been recognized for several decades that globally a major fraction of sulfate is formed in clouds (Roelofs et al., 1998; Barth et al., 2000) and to a smaller extent also in deliquesced aerosol

particles (Sievering et al., 1991; Alexander et al., 2005; Zheng et al., 2015). More recently, it has been shown that also secondary organic aerosol (SOA) mass can be formed by chemical reactions in cloud and aerosol water (aqSOA) (Surratt et al., 2010; Ervens et al., 2011; McNeill, 2015; Xu et al., 2015; Marais et al., 2016). SOA formation in clouds is not always observed, however. For example, Wagner et al. (2015) systematically analysed vertical profiles in the SOA-dominated Southeast US, and found that SOA formation in the fair-weather cumulus clouds was statistically

insignificant.

The formation processes, precursors and conditions for aqSOA formation are more poorly quantified than for sulfate. Tracer compounds such as oxalic acid that are dominant products of aqueous phase processes have been identified but usually only contribute a few percent to the total aerosol mass (Shen et al., 2012; Wang et al., 2012) and, thus, they do not reveal the general role of aqSOA formation to modify aerosol properties. In addition, oxalate might have

additional, less dominant emission sources such as biomass burning (Narukawa et al., 1999; Falkovich et al., 2005; Zhang et al., 2017) and other others (Huang and Yu, 2007). Bulk properties of OA have been shown to be modified differently by aqueous phase processes than by surface or gas phase reactions. These properties include the oxygen-to-carbon (O/C) ratio which is often higher in aqueous-phase derived products (Ervens et al., 2011; Sorooshian et al., 2011; Waxman et al., 2013; Chakraborty et al., 2016) or hygroscopicity (Shingler et al., 2016). However, pathways

that produce aerosol with high O/C via gas-phase reactions are also possible (Chhabra et al., 2011; Ehn et al., 2014; Krechmer et al., 2015), which needs to be taken into account in the interpretation of case studies.

The addition of mass in clouds only occurs on activated particles and often leads to a distinct droplet mode that separates unactivated particles from activated ones (Hoppel et al., 1994). A similar effect of mode separation might be achieved by collision/coalescence within clouds (Feingold et al., 1996); however, these physical processes do not

lead to a distinct change in chemical composition such as the production of aqueous phase tracer compounds. While this size separation might be the most unequivocal microphysical tracer of cloud processing, also the change in bulk and/or size-resolved (physico)chemical properties of the aerosol population might be used to identify cloud-processed aerosol. Not only total aerosol mass, but also its distribution throughout the particle population is important since particle size and composition determine particles' atmospheric lifetime by dry and wet deposition (Maria et al., 2004)

and the aerosol direct and indirect effects on climate (Lin et al., 2014). Thus, it is important to identify and quantify how cloud-derived products affect aerosol loading and properties.

Many current model parameterizations for sulfate and aqSOA formed in clouds apply empirical expressions to



distribute mass throughout the aerosol size distributions (Ervens, 2015, and references therein). However, such approaches might lead to inaccurate representation of time-dependent mass evolution and consequently of particle
lifetime.

In the current study, we apply a combination of model simulations and observations to explore a possible signature of cloud processing in aerosol. Unlike other studies that focused on single parameters, such as modification of size distribution (Eck et al., 2012), aqSOA tracer compounds (Kawamura and Ikushima, 1993; Kawamura and Yasui, 2005; Agarwal et al., 2010), and O/C ratio (Chakraborty et al., 2016), we compare all of these properties in different
air masses, from very clean (background air) to heavily polluted (biomass burning) as they were identified using highly instrumented aircraft data from SEAC⁴RS. Trends in model results are compared to those from other observational data sets, in order to draw conclusions on a possible cloud processing signature in different air mass types.

## 2   Data Sets

### 2.1   Data Sets and Air Masses

#### 2.1.1   SEAC⁴RS (Studies of Emissions and Atmospheric Composition, Clouds and Climate Coupling by Regional Surveys) aircraft field study results as input to model

A rich set of airborne data collected on the NASA DC-8 is used from SEAC⁴RS based out of Houston, Texas in August-September 2013 (Toon et al., 2016). SEAC⁴RS was a multi-platform field campaign addressing issues associated with atmospheric composition over North America. It included two test flights and 21 research flights with
the DC-8 covering altitudes from the surface to above 10 km. Data from this campaign were used to initialize model simulations that will be discussed subsequently.

Owing to the broad range of conditions sampled, a variety of criteria used by Shingler et al. (2016) for the same dataset, were applied to define the following air mass types:

- Biomass Burning (BB) – Wildfire: Acetonitrile > 250 pptv or (when acetonitrile unavailable) CO > 250 ppbv
95        in non-urban areas;

- Biomass Burning – Agricultural: Same as BB - Wildfire, with additional visual confirmation;

- Biogenic: Isoprene + monoterpenes + methyl-vinyl-ketone (MVK) + methacrolein (MACR) > 2 ppbv and acetonitrile < 250 pptv;

- Marine: in planetary boundary layer (PBL), over ocean, and more than 40 km from the coast;

- Urban: in PBL; spatially over Houston (30.50°N, - 94.60°W to 29.00°N, -96.10°W) or Los Angeles (34.17°N, -117.00°W to 33.44°N, -119.75°W);

- Background/Mixed: in PBL; did not fit into first five categories.

Relevant instruments used to obtain data to apply the criteria above included a Proton-Transfer-Reaction Mass Spectrometer (PTR-MS) (de Gouw and Warneke, 2007) for selected species, including MACR, MVK, monoterpenes,
isoprene, and acetonitrile. Isoprene levels in biomass burning plumes represented upper limits owing to interferences from other species such as furan. Furthermore, isoprene levels are higher in the Biomass Burning – Agricultural



category as compared to the Biomass Burning – Wildfire category owing to the aircraft having sampled the former much closer to its source as compared to the more aged plumes of the latter. The MVK and MACR data also are vulnerable to an interference (ISOPOOH), which is most relevant in lower NO regions. As the corresponding NO and

$NO_2$ levels (measured by NOAA $NO_yO_3$ instrument) in **Table S1** are not very low, this potential interference is considered to be minor, except for the biogenic cases, where it is known to be substantial. Data for CO were obtained from a folded-path, differential absorption mid-IR diode laser spectrometer (Sachse et al., 1987). Water vapour data were used from the Diode Laser Hygrometer to identify the height of the PBL (Diskin et al., 2002). **Table S1** also lists the following gases: HCHO (NASA In-Situ Airborne Formaldehyde (ISAF) instrument), $SO_2$ (Georgia Tech Chemical

Ionization Mass Spectrometer (CIMS); Kim et al. (2007)), $H_2O_2$ (Caltech CIMS) and $O_3$ (NOAA $NO_yO_3$ instrument).

An Aerodyne High-Resolution Time-of-Flight Aerosol Mass Spectrometer (HR-AMS) (DeCarlo et al., 2006; Canagaratna et al., 2007; Dunlea et al., 2009) was used for non-refractory composition of submicrometer particles, including the O/C ratio of OA. Given that the HR-AMS, as operated and analysed for SEAC4RS, did not quantify refractory and semi-refractory species, submicron sodium chloride and nitrate in the marine BL are not included in

the AMS results.

Black carbon (BC) data were obtained with a Humidified-Dual Single-Particle Soot Photometer (HD-SP2) (Schwarz et al., 2015). Aerosol size distribution data were used from the Langley Aerosol Research Group Experiment (LARGE) instrument package from a Scanning Mobility Particle Sizer (SMPS; TSI, Inc, model 3080/3010; mid-point $D_p$ between 11 and 316 nm) and an Ultra-High Sensitivity Aerosol Spectrometer (UHSAS; Droplet Measurement

Technologies, Inc.; mid-point $D_p$ between 63 nm and 891 nm). Sizing calibrations were performed frequently during the measurement period using polystyrene latex spheres and monodisperse ammonium sulfate particles for the SMPS and UHSAS, respectively. The two distributions were stitched together at the upper diameter bound of the SMPS, above which the UHSAS data were used (cf **Section 3.2**).

### 2.1.2   Data sets for identifying cloud processing

Data are analysed from several other campaigns. More specifically, the following datasets are used:

(i) Water-soluble anions and cations from a particle-into-liquid sampler coupled to off-line ion chromatography (PILS-IC, Brechtel Mfg. Inc.; (Sorooshian et al., 2006b) deployed on the Center for Interdisciplinary Remotely Piloted Aircraft Studies (CIRPAS) Twin Otter during the Gulf of Mexico Atmospheric Composition and Climate Study (GoMACCS) mission between August and September 2006, based in Houston, Texas;

(ii) Size-resolved aerosol composition from a micro-orifice uniform deposit impactor (MOUDI, MSP Corporation (Marple et al., 1991)) at three ground sites in Arizona (Hayden, Tucson, Mt. Lemmon; (Sorooshian et al., 2012; Youn et al., 2015) and in Marina, California (Maudlin et al., 2015; Braun et al., 2017); and

(iii) Size-resolved aerosol hygroscopic growth factors as measured by a humidified tandem differential mobility analyzer (HTDMA; Brechtel Manufacturing Inc. (BMI) Model 3002; (Wonaschütz et al., 2013)) for samples collected

at a ground site at Mt. Lemmon in Arizona.



## 2.2 Model

### 2.2.1 Model Description

A parcel model is used to simulate cloud-processing in a transect of an air parcel along a prescribed trajectory through a cloud (Feingold and Kreidenweis, 2000; Ervens et al., 2004). Gas phase chemistry occurs during the full simulation;

the chemical scheme is based on the NCAR Master mechanism (Kim et al., 2012). Gas phase precursors for aqSOA include isoprene, toluene, xylene, and ethylene whose oxidation products (glyoxal and related compounds) are taken up into the aqueous phase and further oxidized (Ervens et al., 2004). These precursor compounds and $SO_2$ are not replenished during the simulation, in order to simulate emissions into a cloud away from emission sources. AqSOA formation from these compounds and sulfate formation by $SO_2$ oxidation with $H_2O_2$ and $O_3$ have been described

previously (Ervens et al., 2014; McVay and Ervens, 2017). Aerosol mass formation outside clouds or on/in interstitial particles inside the clouds is not included to focus only on aerosol modification due to aqueous phase processes. We do not include non-oxidative aqSOA formation pathways in our model (e.g. IEPOX formation) as they have been shown to (i) occur on longer time scales and (ii) are most effective in wet aerosol as compared to cloud droplets (Woo and McNeill, 2015). They likely occur on longer time scales than the rapid oxidation reactions. Thus, overall the

predicted total aqSOA mass might represent an underestimate while the formation rate might be overestimated. Particle growth is assumed to only occur via chemical mass addition; it is assumed here that in stratocumulus clouds collision/coalescence processes do not greatly contribute to a change in mass and composition.

It should be noted that we do not aim to reproduce observational results but rather seek trends in aerosol properties (O/C ratio, κ, mass) over a wide range of conditions in different air masses. These values change over the course of a

simulation due to the addition of sulfate and aqSOA mass (mass and κ) and individual aqSOA constituents (O/C ratio). The hygroscopicity parameter κ is calculated as a volume-weighted value of the individual aerosol fractions ($\kappa_{sulf}$ = 0.7; $\kappa_{aqSOA}$ = 0.5, $\kappa_{NH4}$ = 0.6, $\kappa_{NO3}$ = 0.55, $\kappa_{org}$ = 0.1). $\kappa_{org}$ refers to the initial organic aerosol fraction before cloud processing; this organic fraction is likely composed of both SOA and primary organic aerosol (POA). The organics that are added by chemical reactions in the cloud water are referred to here as aqSOA. The hygroscopicity parameter

for this added aqSOA mass ($\kappa_{aqSOA}$) is assumed to be the upper range of oxalate salts (Drozd et al., 2014) since oxalate/oxalic acid is one of the major constituents of cloud aqSOA (Ervens et al., 2011). Using this high value for aqSOA, seems an appropriate assumption to represent the hygroscopicity of organics in their dissolved state, i.e. in cloud water. The initial bulk κ values for each air mass are estimated based on the volume fraction of each species (sulfate, nitrate, organics, chloride, ammonium, black carbon, **Table 1**) using the ZSR approximation (Petters and

Kreidenweis, 2007). The change in κ during the course of the simulation is calculated after each model time step (1 s), i.e. by taking into account the newly formed sulfate and aqSOA masses. The O/C ratio is molecular-based and its evolution is calculated according to aqSOA products (O/C(glyoxal) = 1; O/C(glyoxylic acid) = 1.5, O/C(oxalic acid) = 2, etc) that are added to the initial organic fraction. The bulk ratio is calculated by summing the total number of oxygen atoms across all organic species and dividing by the total number of carbon atoms. The evolution of particle

sizes in 30 size classes (11 nm < aerodynamic diameter < 860 nm) is tracked.





### 2.2.2 Model simulations

The model simulations are initialized with air-mass-specific aerosol and gas-phase compositions for the six air masses. The cases differ in their initial concentrations of aerosol mass, mass fractions, particle number concentrations and initial κ, O/C ratio (***Table 1***) and gas phase mixing ratios (***Table S1***). It is assumed that the aerosol is internally mixed

and particles of all sizes have the same composition, which is supported by the AMS size distributions for most cases, except some fresh biogenic plumes. For simplicity, we consider the same trajectory for all air masses in the parcel model where an air parcel spends about 40 min of the 1-h long simulation time in the cloud (between ~450 s and 1100 s and 1800 – 3600 s, respectively). Cloud chemistry occurs only when a minimum total liquid water content, LWC > 0.01 g m$^{-3}$, is exceeded. When RH drops below 100% cloud droplets evaporate, together with some volatile organics.

Even though organic acids (glyoxylic, oxalic, pyruvic) have relatively high vapour pressures, it is assumed that they stay in the particle phase as they contribute to aqSOA in form of salts and complexes. The low pH value of aerosl water as observed during SEAC$^4$RS might also lead to evaporation of organic acids with low pK$_a$ values. However, the fact that these acids are present in aerosol found during the campaign suggests a complex set of equilibria of acid gas/condensed phase partitioning and salt and complex formation of partially dissolved carboxylates. However, also

formation pathways may be missing from our model studies so that the production rate of organic acids, e.g. the decay of oligomers to organic acids (Lim et al., 2015), is underestimated.

All initial size distributions show already some evidence of cloud processing (***Section 3.2***); however, our model exercise intends to show the extent to which such somewhat aged aerosol populations will be further altered due to aqueous phase processing. In the following, we discuss both the modification of bulk and size-resolved parameters.

## 3 Results and Discussion

### 3.1 Bulk Parameters

#### 3.1.1 Mass increase

The red and green lines in ***Figure 1*** show the predicted increase of sulfate and aqSOA mass (left axis), respectively, over the course of the 1-hr cloud simulations. In all cases, sulfate increases very rapidly and stays constant after all

SO$_2$ has been consumed. The increase in aqSOA is slower as it is formed in multiple oxidation steps from precursors (e.g. isoprene → glyoxal → glyoxylic acid → oxalic acid). The in-cloud time is marked by the black lines at the top of each panel in ***Figure 1***. In the time period between the two cloud passages (~1100 – 1750 s), the lines are horizontal as no mass is added during that time and thus the aerosol mass and properties remain unchanged.

These different time scales are in agreement with previous findings on the comparison of sulfate vs OH-initiated

aqSOA formation in clouds (Ervens et al., 2004) and aqueous aerosol (El-Sayed et al., 2015). Rapid SO$_2$ depletion within clouds and fogs has been observed previously (Husain et al., 2000; Reilly et al., 2001). Our predicted sulfate formation rates (~10$^{-8}$ – 10$^{-5}$ M s$^{-1}$, depending on the air mass) are in general agreement with those as found at Mount Tai, China, for moderately acidic and neutral cloud water (Shen et al., 2012).

The relative proportions of sulfate and aqSOA to total in-cloud mass addition depend on the air mass: While there is





clearly more aqSOA than sulfate in the biomass burning and biogenic scenarios (*Figure 1c, d, and f*), the mass formed
in cloud in the urban and marine scenarios are more sulfate-dominated, even though the total $SO_2$ in these scenarios
is less than the total VOC mixing ratios (cf *Table S1*). While $SO_2$ is completely converted to sulfate (other sulfate
precursors are not considered) on a short time scale, aqSOA mass yields from in-cloud VOC oxidation are much lower
($\leq$ ~10%) as in each oxidation step, volatile compounds are formed (e.g. HCHO, $CO_2$) that do not contribute to aqSOA

(Ervens et al., 2008) but evaporate from the droplets.

The absolute mass increase seems quite large (0.2 - 3 $\mu$g m$^{-3}$ depending on air mass type). However, it should be
remembered that our predictions might exaggerate real conditions as neither physical (deposition) nor chemical
(oxidation of organics to volatile compounds in cloud water) sinks for aerosol mass are included in our model in order
to tease out the clearest signature of aqueous phase processing possible. The predicted increase in sulfate in clouds

has been observed in many previous studies for a variety of air masses, e.g. (Table 1 in Ervens, 2015). Cloud residues
at Mount Tai, China, exhibited large fractions of sulfate and water-soluble organics (Li et al., 2011). In the latter study,
both sulfate and aqSOA were found internally mixed in the droplet mode, which suggests that both were formed in
clouds. Increasing organic mass with altitude in clouds (which can be considered being analogous to processing time)
have been observed in several previous field studies focused on marine and urban air masses, e.g., (Sorooshian et al.,

2007a; Wonaschuetz et al., 2012). *Table 2* compares the distribution of individual organic acids to the total organic
acid content in various air masses, as measured by the PILS-IC method on the CIRPAS Twin Otter during the 2006
GoMACCS campaign. AqSOA tracer compounds such as oxalate, and its main aqueous precursor glyoxylate, are
clearly dominant in clouds whereas in the free troposphere organic acids dominate that significantly originate from
clouds. The comparison of the oxalic acid contributions below, in and above cloud suggests that these three air masses

were connected and mass was transported vertically while it was processed in cloud leading to an increasing oxalate
fraction. Often times, air above clouds might not be cloud-processed but transported horizontally, which may lead to
erroneous interpretation of the role of cloud processing (cf *Section 4*). While the increase in oxalate clearly points to
in-cloud mass formation due to aqueous phase processes, in-cloud aerosol measurements are likely associated with
some uncertainties, in particular due to difficulties of sampling cloud droplets vs interstitial particles, and issues

associated with cloud droplet impacts on inlets, e.g. (Murphy et al., 2004). In contrast to the GoMACCS and other
measurements, no clear aqSOA signature was observed in an SOA-rich biogenic region (Wagner et al., 2015).

### 3.1.2    Changes in bulk hygroscopicity

The initial $\kappa$ is in all air masses lower than that of sulfate ($\kappa_{SO4} = 0.7$) and in most cases even lower than that assumed
for aqSOA ($\kappa_{aqSOA} = 0.5$) (*Section 2.2.1* and *Table 1*). The predicted $\kappa$ values (blue lines in *Figure 1,* first right axis)

increase immediately due to the rapid sulfate addition and then drop when aqSOA mass is added to the processed
particles, corresponding to the changes in absolute masses and mass ratios (*Section 3.1.1*). It should be remembered
that the simulations are set up such that they represent the decay of precursor gases without any replenishment during
the simulation time. If mixing of additional gas-phase precursors occurred continuously into the cloud, the changes in
$\kappa$ might not be as temporally resolved as predicted in *Figure 1.* In such a case, the distinct temporal changes in $\kappa$ due

to sulfate and aqSOA addition, respectively, might be more obscured, e.g. when other secondary organics are added



simultaneously to the particles. The time scales of aqSOA formation might be different for other aqSOA formation processes, i.e. those that are not initiated by the OH radical which would also change the slope of the mass and κ evolution in *Figure 1*. In addition, in the ambient atmosphere the predicted trends in κ could be additionally obscured due to mixing with other air masses into the cloud. The changes of κ in the biomass burning cases (*Figure 1c* and *d*) are overall very small (Δκ ≤ 0.1) except during the sharp peak at the beginning when aqSOA ($\kappa_{aqSOA}$ = 0.5) is added.

### 3.1.3    Changes in bulk O/C ratio

The oxygen-to-carbon (O/C) ratio only reflects the composition of the organic portion of the aerosol (OA). The orange lines in *Figure 1* (second right axis) show the predicted bulk O/C ratios as calculated based on predicted aqSOA formation (*Section 2*). In all cases, the O/C ratio increases close to the beginning of the simulations, with the lines following the same trends as predicted for the aqSOA mass increase (*Section 3.1.1*). The increase in O/C ratio continues still in the second passage of the parcel through the cloud. This increase is caused both by oxidation of dissolved VOCs and by the further oxidation of aqSOA products that have been formed in the cloud water (e.g. oxidation of glyoxylic to oxalic acid). Similar to the findings for the mass increase and κ, the largest increase in O/C ratio can be seen for the urban, marine and biogenic cases with Δ(O/C) ≤ 0.4. For the biomass burning cases, the changes are rather subtle with an increase of Δ(O/C) ~ 0.1. It should be noted that this change might represent an overestimate as we neglect numerous physical and chemical processes that could lead to a weaker increase in Δ(O/C) or even to its decrease. Such processes include non-oxidative reactions that lead to aqSOA which will produce less-oxygenated aerosol. It was discussed that IEPOX might contribute significantly to aqSOA in wet aerosol (Budisulistiorini et al., 2017) or non-photochemical processes occur in fog (Sullivan et al., 2016). Wet deposition or further oxidation of oxygenated and highly soluble aqSOA constituents might lead to a removal of highly soluble organics and thus to an overall decrease of the bulk O/C ratio.

There are not many studies that focus on modifications of the O/C ratio in the aqueous phase. Gilardoni et al. (2016) found an increase in O/C ratio of Δ(O/C) ~ 0.2 upon fog processing in a biomass burning plume. The increase upon processing was similar to the predicted one with Δ(O/C) ~ 0.2 (*Figure 1c, d*). During the Whistler Aerosol and Cloud Study (WACS 2010), Lee et al. (2012) found that in a biogenically-influenced background site, the O/C ratio was clearly enhanced upon cloud processing, similar to the O/C ranges as shown in *Figure 1e and f*. However, they pointed out uncertainties in translating the $f_{44}$ signal from unit-mass resolution AMS measurements into O/C ratio, as the relationships determined by Aiken et al (2008) and Canagaratna et al. (2015)might not be generally valid for all species and ranges of O/C ratios. The largest change in κ and O/C ratio is predicted for the urban, marine and biogenic air masses (*Figure 1a, b, f*) whereas the changes in the biomass burning cases (*Figure 1c, d*) and background air (*Figure 1e*) are smaller. B

### 3.1.4    The ratio of potential added mass from precursor gases to initial aerosol mass

Biomass burning scenarios are characterized by high aerosol mass loadings and high - mostly organic – trace gas mixing ratios. The background/mixed case has a higher initial mass (3.86 μg m$^{-3}$, *Table 1*) than the marine, urban and





biogenic air masses whereas the precursor gases are approximately on the same order of magnitude as in these three
air mass types (**Table S1**). In order to significantly change the bulk properties of initial aerosol by additional $SO_2$ and
aqSOA mass, the newly-formed mass has to comprise a substantial fraction of the total mass so that the volume-based
κ and the molecular-based O/C ratio are significantly changed. Based on this idea, we calculate an initial potential
added mass due to aqueous processing of precursors-to-pre-existing mass ratio for each air mass:

$$R_{tot} = \underbrace{\frac{[SO_2]\cdot 98/64}{m_0}}_{R_{SO4}} + \underbrace{\frac{Y\cdot[aqSOA\ prec]}{m_0}}_{R_{aqSOA}}\ [\mu g\ m^{-3}/\mu g\ m^{-3}]$$
Equation-1

Where $[SO_2]$ is the mass concentration of $SO_2$ [μg m$^{-3}$], the factor of 98/64 accounts for the mass difference of $H_2SO_4$
vs. $SO_2$, and [aqSOA prec] is the total mass concentration [μg m$^{-3}$] of all VOCs that may act as precursors for aqSOA
(**Table S1**). These precursors include isoprene, methyl vinyl ketone, methacrolein, toluene, xylene and ethylene. The
numerator is the potentially added mass, i.e. the mass that would be added to the initial mass $m_0$ [μg m$^{-3}$] due to cloud
processing, if the precursors were completely consumed. The VOC mixing ratio is multiplied with an approximate
effective mass yield factor Y, in order to account for the facts that (i) only part of the VOCs mass will be converted
into aqSOA and (ii) aqSOA species can be further oxidized to $CO_2$ and thus – unlike sulfate – it is not a preserved
mass. An effective yield of 10% is assumed in the remainder of this study, based on model studies that have shown
that the aqSOA yield from isoprene is at most 10%, depending on cloud and $NO_x$ conditions (Ervens et al., 2008).
This previous model study might not have included all aqSOA precursors and formation pathways so that the mass
yields reported there might represent an underestimate. If more updated information on yields for specific precursors
becomes available, the value of Y used in Equation 1 can be updated accordingly.

In **Table 3**, the R values for all six air masses are listed. The highest value ($R_{tot}$ = 5.2) is shown for the marine scenario
followed by the values for the biogenic ($R_{tot}$ = 0.2.0) and urban (Rtot = 2.0) cases. The lowest values are shown for
the biomass burning cases (R = 0.012 and R = 0.56 for the wildfire and agricultural burning, respectively) with a
similar value for the background case (R = 0.72). While the VOC mixing ratios in the biomass burning air masses are
relatively high, the inefficient conversion into aerosol mass (as compared to sulfate) and the high pre-existing aerosols
lead to an overall low R value. This is in agreement with previous studies that showed that in biomass burning plumes
large fractions of organic material reside in the particle phase as compared to the gas phase (Heald et al., 2008; Cubison
et al., 2011). Even though the agricultural biomass burning air mass contains the highest $SO_2$ mixing ratio among all
six air masses, the added sulfate is not sufficient to alter the properties of the initial aerosol mass $m_0$, which is also the
highest among all cases (**Table 1**). All $R_{SO4}$ values are higher than the $R_{aqSOA}$ values for the same air mass. This trend
suggests that generally the addition of sulfate to an initial aerosol population might more efficiently change the initial
aerosol population than the addition of aqSOA. The trends in **Table 3** give some guidance for which air masses a
cloud-processing signature may be expected: The higher the ratio R, the more susceptible the pre-existing aerosol
mass is to be substantially enhanced by in-cloud mass formation.

Air masses in the Southeast US are usually categorized as biogenic, but yet only little evidence of cloud processing
was observed which was mostly ascribed to sulfate addition (Wagner et al., 2015). Applying the concept of the ratio





R to these air masses, it can be shown that the initial mass of ~10 µg m$^{-3}$ was relatively high whereas the precursor

concentrations were comparably low ([SO$_2$] ~ 0.3 ppb, [Isoprene] ~ 1.5 ppb, [Aromatics] < 1 ppb) (Lu et al., 2015; Wagner et al., 2015), resulting in R$_{SO4}$ = 0.5, R$_{aqSOA}$ = 0.008 and R$_{tot}$ = 0.6, respectively.

In contrast, above Houston, cloud processing was observed (Wonaschuetz et al., 2012). The air masses there contained lower aerosol mass but higher precursors (m$_0$ ~ 5 µg m$^{-3}$; [SO$_2$] = 1.5 ppb; [Aromatics] ~ 8 ppb, [Isoprene+MVK+MACR] ~ 4 ppb), yielding R$_{SO4}$ = 0.5, R$_{aqSOA}$ = 0.8 and R$_{tot}$ = 1.3, respectively, slightly lower the

result for the urban air mass in the current model study (**Table 3**).

### 3.2    Changes in size-resolved parameters

Bulk properties do not allow any detailed conclusions about the effect of cloud processing on individual particles and, thus, on their resulting composition and size. While being more complex both in terms of measurements and model simulations, only size-resolved measurements and model studies permit such conclusions and are discussed

subsequently. Chemical processes in cloud droplets might significantly change the properties of the droplet residuals. Depending on the activated fraction, this modification might change the bulk properties of the total aerosol population to different extents. It has been discussed by Ervens et al. (2014) that in small droplets, i.e. in particles with a relatively high surface-to-volume ratio, more efficient aqSOA formation can be expected as oxidation rates might be enhanced due to efficient oxidant and precursor uptake. The size of cloud droplets is not a strong function of the size and/or

composition of the CCN but it is mostly determined by the growth history and competition for water vapour within the cloud. As size-resolved composition measurements from SEAC$^4$RS are very noisy, any conclusions based on this data might be inconclusive.

### 3.2.1    Size-resolved mass increase

Many studies have discussed the formation of a droplet mode upon cloud processing, which is caused by mass addition

to activated particles only. Such predicted evolution of the aerosol size distribution is shown for the six air masses in **Figure 2**. The black symbols and lines denote the aerosol size distribution that was used as model input (black symbols are mostly covered by colored symbols).

Upon cloud processing, the predicted separation of the cloud-processed particles from the smaller-sized particles is different in the six air masses. Processed size distributions in **Figure 1** are overlaid on the initial size distributions and

color-coded by the relative mass increase, i.e.

$$\text{Relative mass increase [\%]} = \left(\frac{\text{Mass after cloud processing}}{\text{Initial mass}} - 1\right) \cdot 100\% \qquad \text{Equation -2}$$

This relative mass increase is similar to the parameter R as defined in **Equation 1** in the sense that it shows the resulting mass increase upon processing of the precursors after cloud processing. In agreement with the trends as identified for the bulk masses (Section 3.1.1), the two biomass burning scenarios show the smallest relative mass increase with

largest values of ~10% and 17%, respectively. In all other cases, the mass of some particles might double (relative mass increase ~100%). **Table 4** summarizes the maximum relative mass increase for individual sizes, together with



the particle size range that is mostly affected by cloud processing. In the two biomass burning scenarios, only particles with diameters > ~250 nm show any processing. Due to the high particle number concentration in these cases, the maximum cloud supersaturation is suppressed because the numerous particles act as an efficient condensation sink for

water vapour. Consequently, only a small fraction of the aerosol population is activated into cloud droplets. This small activated fraction explains the rather small changes in bulk κ and O/C ratio (***Figure 1***). Cloud processing often leads to the separation of unactivated and activated particles within the aerosol size distribution due to the 'Hoppel minimum'. In ***Figure 2a, b, e, f*** the particles around 100 nm are affected most strongly and also show some sign of this separation into a droplet mode. Thus, it is predicted that cloud processing leads to a shift to larger particle sizes

and a narrowing of the size distribution (Feingold and Kreidenweis, 2000). In the biomass burning cases (***Figure 2c, d***), the most affected particles are near the maximum of the main size mode and the shift to larger sizes is not as clear.

The absolute mass increase is in all cases around several ng m$^{-3}$ in the individual size classes that are separated by d(log bin-width) ~ 0.05 (***Figure S1***). Whereas this translates into doubling of particle mass in the cleaner air masses, the relative mass increase in the biomass burning cases is much smaller owing to the initial high particle loading, i.e.

low R values (***Equation 1***). Tracers of aqueous phase processing have been identified, e.g. by (Cook et al., 2017), in cloud samples that were affected by biomass burning plumes. However, such analyses do not reveal the extent to which the total aerosol population might have been altered in the cloud. Our results show that they only represent a small fraction of the total aerosol population.

Many previous studies have identified a droplet mode upon cloud processing. An overview article has been given by

(Eck et al., 2012). In ***Figure 3***, exemplary results of cloud processing in urban, marine and remote air masses are shown. The relative mass increase of sulfate and oxalate is examined as tracers for cloud processing. For two sites, data were compared between a moist and dry period; more specifically, data were compared between monsoon months (July – September) and a dry period (June) for the urban area in Tucson, Arizona, and also between a monsoon period and a drier period in November for a remote site in Hayden, Arizona. Finally, data were compared between a fire

period and a non-fire period in Marina, California during the summer when there is persistent cloud coverage. A consistent feature for the two Arizona sites was that a peak in the relative mass increase (Monsoon versus other periods) for sulfate and oxalate was between 0.32 - 0.56 µm, which is consistent with the droplet mode. While the fire and non-fire comparison does not contrast periods with varying moisture levels, it contrasts periods with varying amounts of precursors that still reveal the importance of aqueous processing in terms of the greater mass production

when precursors are more plentiful. The relative mass increase for the comparison of fire and non-fire conditions in the coastal/marine area with persistent cloud coverage was highest between 0.56-1 µm, in agreement with the larger critical diameter (i.e., smaller activated fraction) in ***Figure 2***. Similarly, analysis of fog-processed aerosol in Fresno, California, also revealed a clear signature in terms of size distribution and composition changes (Ge et al., 2012). In this latter study, both sulfate and aqSOA accumulated at particle sizes above ~200 nm upon cloud processing.

**3.2.2    Changes in size-resolved hygroscopicity (κ)**

***Figure 4*** shows the same parcel model results as in ***Figure 32,*** but color-coded by κ instead of the relative mass





increase. Unlike *Figure 1* that shows the time evolution of κ, in *Figure 4*, the model-predicted values after one hour of processing are shown. Conclusions are similar to those in *Section 3.1.2* as the smallest changes in κ are seen in the biomass burning cases where only a small fraction of the aerosol population is processed (*Table 4*) and the high initial
mass is not increased substantially by the addition of sulfate and aqSOA.

The hygroscopic growth factor g(RH) is a measure of particle hygroscopicity. HTDMA measurements of initial and cloud-processed aerosol can give evidence of cloud-processing as more sulfate and aqSOA mass is added to larger particle sizes, enhancing the hygroscopicity of previously less hygroscopic particles. *Figure 5* shows an example of a size distribution of aerosol hygroscopic growth (shown as κ) atop Mt. Lemmon in Arizona in relation to chemical
mass fractions of selected water-soluble species including inorganic and organic acid ions. The size ranges with the highest κ values exhibit the highest mass fractions of sulfate. The results show how it is difficult to isolate the impact of aqSOA. Although the contribution of oxalate to the total organic acid mass is highest for diameters in the range of 0.32-0.55 µm (41%), that same stage exhibited the highest sulfate mass fraction (73%) and inorganic mass fraction (88%). This may have trumped the smaller effect of a change in the functionality of the organic fraction of the aerosol.
In a separate study in the marine boundary layer off the California coast, Hersey et al. (2009) measured reduced size-resolved aerosol hygroscopic growth factors above the stratocumulus cloud top as compared to the sub-cloud region as a result of enhanced bulk aerosol organic mass fractions above cloud. However, the air masses were different below and above cloud, with continental free tropospheric air enriched with organics residing above cloud top and more inorganic-rich aerosol below cloud bases. This observation demonstrates that comparisons of below- and above-cloud
air should be performed carefully as differences in aerosol properties are not necessarily due to cloud processing. A recent study comparing inflow and outflow aerosol from deep convective storms revealed that although size-resolved κ values may not have exhibited a significant enhancement in the anvil outflows (and sometimes reduced values), the signature of aqueous processing could have been missed as a result of lateral entrainment and mixing of less hygroscopic aerosol mixing with the processed aerosol that entered at the storm cloud base (Sorooshian et al., 2017);
this might be also a consequence of different scavenging efficiencies of sulfate and organics, respectively (Yang et al., 2015). A case was profiled where biomass burning aerosol, with low hygrosopicity, entrained into a storm and resulted in a lower mean κ value in the outflow as compared to the inflow. An altitude-dependent entrainment model was applied to their analysis to show that the measured κ value exceeded that predicted for the outflow, revealing that a process, most likely aqueous processing, helped increase the hygroscopicity of the aerosol.

**3.2.3    Changes in size-resolved O/C ratio**

The same figure as for mass increase and κ change is once more reproduced in *Figure S2* showing the change in O/C ratio throughout the aerosol distribution upon cloud processing. In the marine case (*Figure S2a*), the O/C ratio is predicted to increase by about 0.5 units in the activated fraction. As the smallest activated particles are smaller than 100 nm and thus the activated fraction is substantial, this change in O/C ratio is also reflected in the bulk O/C ratio in
*Figure 1* and translates into a high R value (Equation 1). Changes in the O/C ratio are rather small in the urban case, as the strong increase in mass is mostly due to sulfate which does not affect the O/C ratio. It can be expected that in biomass burning scenarios, cloud water might contain highly oxidized organics, and thus a high O/C ratio (Gilardoni



et al., 2016; Cook et al., 2017). However, as the dissolved mass only comprises a small fraction of the total particle number, this oxidation might not affect bulk aerosol properties to a large extent.

Fog water analysis in the Indo-Gangetic plains revealed higher O/C in small fog droplets with a difference of Δ(O/C) ~ 0.2 between small and large fog droplets (Chakraborty et al., 2016). Model studies explained this difference with the larger surface-to-volume ratio of smaller droplets, which allows for more efficient uptake of oxidants such as OH and aqSOA precursors from the gas phase (Ervens et al., 2014). As OH is assumed to be (one of) the most efficient oxidants of organics in cloud droplets, the resulting higher OH concentration leads to relatively more aqSOA and a

higher O/C ratio in small droplets. The fog study by Chakraborty et al. (2016) might not be directly comparable to the model results in *Figure S 2* that contrast activated and non-activated particles upon cloud processing. However, the fog studies show that drop size plays an important role for aqSOA formation. Given the high cloud drop number concentration in the biomass burning cases in the SEAC$^4$RS biomass burning scenarios (a few 1000 cm$^{-3}$, as opposed to a few 100 cm$^{-3}$ or less in the other cases), the smallest cloud droplets might be present in these cases. Thus, it can

be expected that the aqSOA formation rates in such cloud droplets are highest (Ervens et al., 2014) due to the favorable total surface-to-volume ratio (McVay and Ervens, 2017). In fact, in both biomass burning scenarios, nearly 10 μg m$^{-3}$ organic mass are added (*Figure 1c and d*) in agreement with observations of efficient aqSOA formation in cloud-processed biomass burning plumes (Gilardoni et al., 2016). However, this mass is not sufficient to change the properties of the pre-existing aerosol mass (*Section 3.1.4*).

**4     Caution in characterizing air near clouds to detect the cloud-processing signature**

        The previous model analysis suggests that detecting unambiguous evidence of a particle having undergone aqueous processing as compared to clear-air processing is challenging in the ambient atmosphere. Evidence of cloud processing strongly depends on the air mass and its history. This is in sharp contrast to controlled laboratory experiments where conditions can be controlled and optimized to detect an aqueous signature by an increase in O/C ratio, κ or formation

of tracer compounds (Lim et al., 2010; Lee et al., 2012). In the ambient atmosphere, several interferences might obscure the signature and/or lead to false conclusions.

        (i) Studies with vertically-resolved measurements below, inside, and above cloud have an added complication that clouds can be decoupled from a significant portion of the sub-cloud layer (Wang et al., 2016), or there can be a very sharp temperature inversion immediately above their cloud top that leads to a different air mass above the tops

associated with the free troposphere (Dadashazar et al., 2018). Thus, continuity in meteorological parameters, such as temperature and/or relative humidity should be carefully taken into account before conclusions are drawn on cloud processing. Aircraft that fly even within 10 m above cloud top in the entrainment interface layer, such as in subtropical stratocumulus regions, still have influence from free tropospheric air masses (Dadashazar et al., 2018). *Figure 6* demonstrates an example of an airborne experiment where particles with higher κ were found above cloud than below

and in cloud. However, this trend is coincidental as the air mass above did not originate from the cloud. In other studies, such an increase in κ was correctly attributed to sulfate addition due to cloud processing (Shingler et al.,



2016). Similar mixing of air masses from the free troposphere and the boundary layer were observed within the inter-cloud layer in the Southeast US (Wagner et al., 2015).

(ii) While likely the majority of atmospheric oxalate is formed in clouds, it has been shown that oxalate might have
additional sources, such as biomass burning (Narukawa et al., 1999; Falkovich et al., 2005; Zhang et al., 2017) or direct emissions (Huang and Yu, 2007). In addition, oxalate and other aqSOA compounds can get further oxidized in clouds, in particular in the presence of iron (Furukawa and Takahashi, 2011; Kawamura et al., 2012; Sorooshian et al., 2013). In addition, oxalic acid and other (weaker) organic acids might evaporate from acidic aerosols. Thus the lack of a clear oxalate increase is not necessarily indicative of processing in cloud-free air only. Finding correlations
between aqueous organic tracer species and sulfate (Yu et al., 2005; Huang et al., 2006) are not necessarily indicative of causality. Co-variance alone (or the lack thereof) of tracer species (e.g., oxalate) with their aqueous precursors such as glyoxal or glyoxylate (e.g., Sorooshian et al., 2006a; Rinaldi et al., 2011) are not a sufficient indicator to conclude on cloud processing.

(iii) Aerosol composition can be altered during sampling due to possible fragmentation and volatilization of
aqSOA products in counterflow virtual impactor (CVI) inlets used to isolate cloud droplet residual particles (Shingler et al., 2012; Prabhakar et al., 2014).

(iv) Although still limited in their ability to provide direct proof, reports of size-resolved field measurements may miss out on the full story of an aqueous signature if only the submicrometer size range (e.g., droplet mode) is examined, as aqueous processing can influence the composition of aerosol in the coarse mode (e.g., Deshmukh et al., 2017).

(v) Not only aqueous phase processing in clouds but also in deliquesced aerosol particles can lead to aerosol mass. While this is rather inefficient for sulfate, many laboratory and ambient studies suggested that aqSOA can be efficiently formed in cloud-free, high relative humidity conditions. Characteristics of this aqSOA mass might be similar to cloud aqSOA (highly oxygenated and functionalized). Correlations of increased aqSOA mass with increasing relative humidity or cloud vs non-cloud scenarios should be interpreted with caution (Youn et al., 2013).
RH is usually higher in the morning, when pollution and the boundary layer thickness and mixing are different than in the afternoon. Similar to findings for nitrate (Lee et al., 2003), it can be expected that the temperature increase during the day might lead to a decrease in organic mass due to the volatilization of semivolatile SOA.

## 5    Conclusions and Implications

We have analysed data sets from the SEAC[4]RS and other field experiments in order to identify various aerosol
properties that might show evidence of aqueous phase processing of aerosol particles within clouds. In total, three properties, namely mass increase, hygroscopicity ($\kappa$) and O/C ratio were explored by means of model studies for six different air masses (urban, marine, wildfire biomass burning, agricultural biomass burning, biogenic and background). Model results suggest that in moderately polluted air masses, such as in urban, marine and biogenic scenarios, changes in particle mass and properties can be most easily identified. In order to quantify the susceptibility
of an aerosol population to be significantly modified by clouds, we define a mass ratio $R_{tot}$, which is the ratio of possible precursor gases for aerosol mass formation ($SO_2$, VOCs), i.e. the potential aerosol mass, and the initial aerosol





mass. The biomass burning cases show the lowest values of $R_{tot}$ (0.12 and 0.56, respectively) whereas the marine air mass is characterized by the highest value of $R_{tot}$ = 5.2. Calculating this ratio for previous experiments in different air masses explains why in some cases (e.g., urban) cloud processing was observed whereas it was not clearly detected

in a clean biogenic scenario. Generally, the mass ratio $R_{SO4}$ (ratio of potential sulfate mass to initial aerosol mass) is larger than the values for $R_{aqSOA}$ (potential aqSOA mass versus initial aerosol mass). Thus, sulfate addition likely leads to more aerosol modification during cloud processing than aqSOA addition. Thus, the O/C ratio might not change significantly due to cloud processing as it only describes the organic aerosol fraction. Other parameters that describe the total aerosol mass, such as mass increase or a change in the hygroscopicity parameter κ might be more useful to

detect a signature of cloud processing.

Mass addition to initial high particle loadings, as encountered in biomass burning plumes, might not be sufficient to modify total mass and physico-chemical properties to a large extent. All else being equal (e.g. vertical velocity, cloud processing time), the activated fractions in clouds in clean air masses (low particle number concentrations) are higher than under polluted conditions (high particle number concentrations). High number concentration of particles in the

biomass burning cases prevents high supersaturations and, thus, only large particles are activated into cloud droplets where processing occurs. As a result, only a small fraction of particles are cloud-processed in biomass burning plumes. As the ratio of activated to total particles is much larger in less polluted air masses, relatively more particles form cloud droplets and undergo addition of sulfate and aqSOA mass.

While the presence of tracer compounds of aqueous phase processing, such as hydroxyl methane sulfonate (Munger

et al., 1986), oxalate (e.g., Huang et al., 2006; Sorooshian et al., 2010; Wonaschuetz et al., 2012) or oligomers (Mazzoleni et al., 2010) have been used to conclude on aqueous processing, these compounds usually only comprise a small fraction of the total aerosol mass and, thus, give only limited quantitative information on the role of aqueous phase processing on the modification of aerosol. Further, tracers like oxalate might have additional chemical sinks, such as to complexation with iron or other trace metals and subsequent photolysis, e.g., Sorooshian et al. (2013).

Therefore, our approach to look at both bulk and size-resolved aerosol properties gives a more comprehensive idea of the role of aqueous phase processes in clouds or wet aerosol on aerosol modification. It should be noted that our model assumptions likely represent overestimates of this signature as we do not include aqueous phase processes that might act as efficient sinks for organic mass. It is likely that water-soluble organic particle constituents (e.g., SOA from sources other than aqueous-phase processes) get oxidized to volatile compounds within cloud droplets and thus the

total SOA mass might decrease whereas aqSOA material is added. In addition, evaporation of organic acids from aerosol particles at low pH might lead to further decrease in aqSOA. The analysis and interpretation of data sets acquired near clouds should be performed with care and it should be made sure that air masses in and above clouds are coupled.

Overall, it can be stated that there is no unambiguous answer to the initial question in the title of this study as to

whether there is a signature of cloud processing on aerosol. The extent to which aerosol properties are modified by chemical processes in clouds depends on the initial aerosol mass, particle number concentration and sulfate and aqSOA precursor gases, as quantified by the mass ratio $R_{tot}$.





Our findings are expected to provide guidance on future field and model studies targeting the role of cloud processing on aerosol properties and total ambient aerosol loading. The lack of a signature does not imply that no aqueous phase

processing occurs. In such cases the signature might have been masked by other processes, which include physical and chemical removal processes of aerosol mass.

### Data and code availability

All data from DC3 and SEAC⁴RS are publicly available from the NASA Langley Research Center Atmospheric

Science Data Center: https://www-air.larc.nasa.gov/missions/dc3-seac4rs/index.html and https://www-air.larc.nasa.gov/missions/seac4rs/, doi:10.5067/Aircraft/SEAC4RS/Aerosol-TraceGas-Cloud, respectively. CIRPAS Twin Otter data can be found elsewhere (Sorooshian et al., 2017b, 2018). Complete model results are available upon request from BE.

### Acknowledgements

AS was funded by Office of Naval Research grant N00014-10-1-0811, N00014-11-1-0783, N00014-10-1-0200, N00014-04-1-0118, and N00014-16-1-2567, and NASA grants NNX12AC10G and NNX14AP75G. TS was supported with a NASA Earth and Space Science Fellowship (NNX14AK79H). PTR-MS measurements during SEAC⁴RS were supported by the Austrian Federal Ministry for Transport, Innovation and Technology (bmvit) through the Austrian Space Applications Programme (ASAP) of the Austrian Research Promotion Agency (FFG). Tomas

Mikoviny is acknowledged for support with data collection and analysis. PCJ and JLJ were supported by NASA grants NNX12AC03G and NNX15AT96G. The authors acknowledge Andreas Beyersdorf, Anne E. Perring, and Joshua P. Schwarz for SEAC⁴RS data used, and they acknowledge several SEAC4RS participants for providing gas-phase date (John Crounse, Alex Teng, $NO_yO_3$ team, Greg Huey, David Tanner, Xiaoxi Liu, and Thomas Hanisco)

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



**Table 1:** Initial aerosol properties for six air masses during SEAC[4]RS. These data are used as inputs to the box and parcel models in order to simulate aqueous phase processing.

| | Marine | Urban | Biomass Burning | Agric Biomass Burning | Background | Biogenic |
|---|---|---|---|---|---|---|
| Relative contributions [%] | | | | | | |
| Ammonium | 10.6 | 8.1 | 2.8 | 2.1 | 7.4 | 7.3 |
| Chloride | 0.6 | 0.1 | 0.3 | 1.9 | 0.8 | 0.2 |
| Nitrate | 1.3 | 1.4 | 3.8 | 2.7 | 2.0 | 1.6 |
| Organics | 21.5 | 55.8 | 89.1 | 90.5 | 64.0 | 63.9 |
| Sulfate | 64.7 | 33.6 | 1.9 | 2.1 | 24.8 | 26.1 |
| Black carbon | 1.3 | 1 | 2 | 0.8 | 1 | 1 |
| $N / cm^{-3}$ | 651 | 5,551 | 3,481 | 9,762 | 3,377 | 2,065 |
| Total mass / $\mu g\ m^{-3\ 1)}$ | 0.33 | 1.25 | 10.5 | 12.1 | 3.86 | 1.74 |
| $\kappa$ | 0.59 | 0.41 | 0.21 | 0.21 | 0.37 | 0.37 |
| O/C | 0.92 | 0.65 | 0.65 | 0.48 | 0.66 | 0.57 |

1) Masses are given in standard $m^{-3}$



**Table 2:** Percent contributions of individual organic acids to total organic acid mass in aerosols in different regions of the lower troposphere during the 2006 GoMACCS campaign, based in Houston, Texas (Sorooshian et al., 2007b)
. "Cloud-CVI" corresponds to droplet residual particle measurements, "Clear Air" is in the boundary layer in cloud-free conditions, and "Free Troposphere" is determined by meteorological sounding profiles.

|  | Below Cloud | Cloud-CVI | Above Cloud | Clear Air | Free Troposphere |
|---|---|---|---|---|---|
| Oxalate | 0.68 | 0.77 | 0.87 | 0.63 | 0.54 |
| Malonate | 0.00 | 0.02 | 0.01 | 0.01 | 0.03 |
| Succinate | 0.02 | 0.01 | 0.00 | 0.01 | 0.02 |
| Glutarate | 0.01 | 0.01 | 0.01 | 0.00 | 0.03 |
| Adipate | 0.00 | 0.00 | 0.00 | 0.00 | 0.02 |
| Suberate | 0.01 | 0.00 | 0.00 | 0.01 | 0.00 |
| Pyruvate | 0.00 | 0.00 | 0.00 | 0.01 | 0.00 |
| Glyoxylate | 0.00 | 0.01 | 0.00 | 0.00 | 0.00 |
| Acetate | 0.10 | 0.13 | 0.07 | 0.16 | 0.19 |
| Formate | 0.09 | 0.03 | 0.02 | 0.06 | 0.04 |
| Benzoate | 0.01 | 0.00 | 0.00 | 0.04 | 0.00 |
| MSA | 0.06 | 0.00 | 0.01 | 0.05 | 0.10 |
| Maleate | 0.01 | 0.01 | 0.00 | 0.01 | 0.03 |





**Table 3:** Mass ratios of potential aerosol mass and initial aerosol mass $m_0$ from $SO_2$ ($R_{SO4}$), aqSOA precursors ($R_{aqSOA}$) and total mass ratios ($R_{tot}$) (Eq-1).

|  | Marine | Urban | Biomass Burning | Agric. Biomass Burning | Back-ground | Biogenic |
|---|---|---|---|---|---|---|
| $SO_2$ [$\mu g\ m^{-3}$] | 1.1 | 1.5 | 0.66 | 3.8 | 1.7 | 1.5 |
| aqSOA precursors [$\mu g\ m^{-3}$] | 0.16 | 1.9 | 2.9 | 9.8 | 1.9 | 11.5 |
| $m_0$ [$\mu g\ m^{-3}$] | 0.33 | 1.25 | 10.5 | 12.1 | 3.86 | 1.74 |
| $R_{SO4}$ | 5.1 | 1.8 | 0.1 | 0.5 | 0.7 | 7.0 |
| $R_{aqSOA}$ | 0.005 | 0.15 | 0.03 | 0.08 | 0.05 | 0.16 |
| $R_{tot}$ | 5.2 | 2.0 | 0.12 | 0.56 | 0.72 | 2.0 |





**Table 4:** Summary of model results describing the aerosol properties upon cloud processing. $\Delta\kappa$, $\Delta m$, $\Delta(O/C)$, and $\Delta Diam$ values denote the largest predicted change as shown in Figures 3, 5, and S1, respectively.

| | Size range [nm] | $\Delta\kappa_{max}$ | $\Delta m(relative)_{max}$ [%] | $\Delta(O/C)_{max}$ | $\Delta Diam_{max}$ [nm] |
|---|---|---|---|---|---|
| **Marine** | 80-300 | 0.05 | 100 | 0.5 | 20 |
| **Urban** | 100-400 | 0.13 | 90 | 0.12 | 37 |
| **BB** | 200-400 | 0.04 | 10 | 0.02 | 18 |
| **Ag BB** | 250-430 | 0.08 | 18 | 0.17 | 18 |
| **Background** | 100-350 | 0.16 | 80 | 0.14 | 45 |
| **Biogenic** | 100-400 | 0.18 | 130 | 0.33 | 50 |





Figure 1: Predicted change in aerosol properties due to cloud processing of six different air masses identified during SEAC[4]RS. Cloud processing simulations are performed for one hour during which a cloud exists for ~ 30 min. Green and red lines show predicted increases in organic and sulfate mass (left axis), respectively; blue and orange lines represent the change in hygroscopicity parameter κ (first right axis) and O/C ratio (second right axis), respectively. The thick black lines near the top of the panels denote the in-cloud time.





**Figure 2:** Predicted relative mass concentration increase due to cloud processing in six air masses as identified during SEAC[4]RS. Black symbols show the measured, initial size distributions; colored symbols are model results, color-coded by relative mass increase (Equation 2)





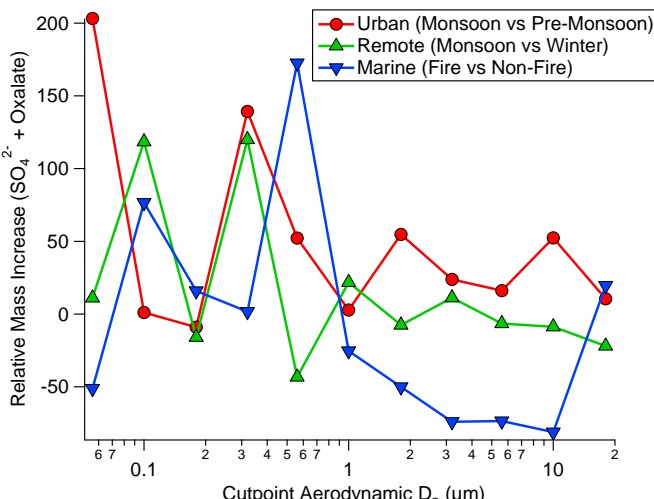

**Figure 3:** Summary of the relative mass concentration increase for sulfate plus oxalate as a function of dry particle size for three scenarios: monsoon – pre-monsoon changes in an urban area (inner city Tucson, Arizona), monsoon – winter changes in a remote area in central Arizona (Hayden, Arizona), fire – non-fire changes in a coastal/marine area with persistent cloud coverage in July-August (Marina, California).



**Figure 4:** Same as Figure 2 but the processed aerosol mass distribution is color-coded by κ



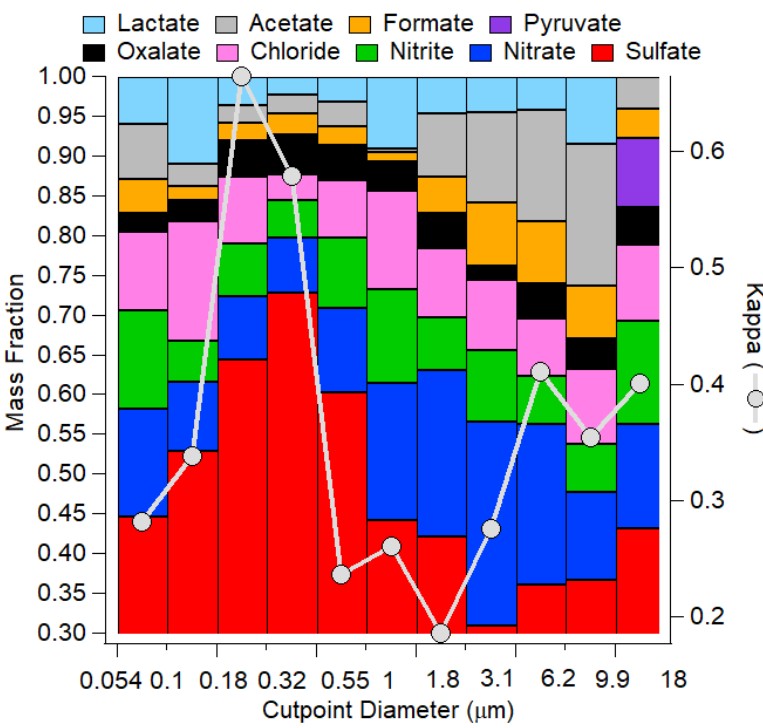

**Figure 5:** Size-resolved aerosol hygroscopic growth (as kappa) and chemical mass fractions as a function of dry particle diameter at a mountaintop site (Mount Lemmon, Arizona; February 2010).





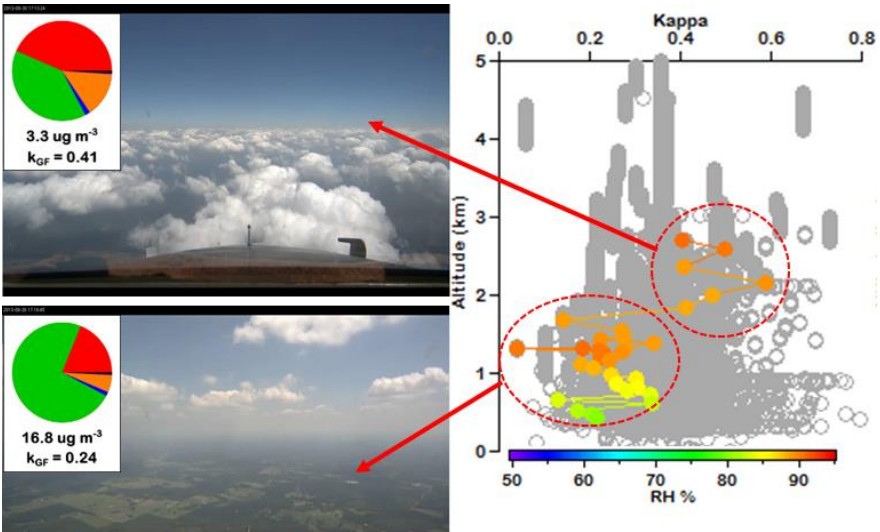

**Figure 6: Left:** Photographs taken from the NASA DC-8 showing where the aircraft was relative to clouds during a SEAC[4]RS flight on 30 August 2013. The pie charts correspond to AMS chemical mass fractions non-refractory aerosol (green = organic; red = sulfate; orange = ammonium; blue = nitrate) and for black carbon (in black) as measured by the HD-SP2 instrument. The average total submicrometer mass concentrations and GF-derived κ values are shown below the pies for above-cloud base and sub-cloud base sampling.
**Right:** Vertical profile of size-resolved GF-derived κ for particles with dry diameters between 180-400 nm with gray being all points during the flight and the colored points being for the specific measurements near the cloud field. While the RH values of the humidified channel of the DASH-SP are shown for each measurement by the cloud field, κ values are shown to allow for a fair comparison regardless of the humidified RH.