# Peer review of "Is there an aerosol signature of chemical cloud processing?"

_Atmospheric Chemistry and Physics, 2018_

## Referee Comment (RC1) · Anonymous Referee #1 · 30 Jun 2018

The manuscript investigates the impact of sulfate and aqueous secondary organic aerosol (aqSOA) formation through cloud processing on relative aerosol mass increase, aerosol hygroscopicity, and organic aerosol oxygen content, focusing first on aerosol bulk properties and then on size segregated properties. Different air mass categories are studied, using measurements collected during the SEAC$^4$RS experiments on board the NASA DC-8, based out of-Houston. Simulations indicated that the impacts of cloud processing are more prominent on polluted air masses than clean background air masses, but less evident in heavily polluted conditions, such as in biomass burning plumes. One of the main implications of this work is that changes in aerosol mass and particle hygroscopicity are better indicator of cloud processing than chemical markers, due to their chemistry sinks. Nevertheless, depending on the initial proper-

ties of air masses, the impact of cloud-processing on the above-mentioned parameters might vary significantly, making sometimes difficult to detect aqueous phase processing, especially under clean conditions or in heavily polluted air masses.

Cloud processing is attracting a growing interest from both the observation and modeling research communities, due to its impacts on air quality and climate. Often field observations struggle to identify aqueous phase processing, which is observed instead during laboratory experiments, delaying its description in chemistry transport models. The present manuscript helps to explain some of the discrepancies among laboratory observations, field observations, and modeling results, and deserve publication in ACP with minor revisions.

General comments: The completeness of data collected during the NASA DC-8 experiments seems to be only partially deployed. For example, the HR-AMS data collected during the flights could be used to characterize the initial O/C and k parameters. For example, the model assumes that $k_{org}$ is equal to 0.1, while Jimenez et al. (2009) shows that, depending on the organic oxygen content the $k_{org}$ can vary from about 0 up to 0.2. Please explain if the use of specific $k_{org}$ for different air mass types could have supported a more accurate analysis and justify why the authors decided to use a constant $k_{org}$ for different masses.

Some authors observed that aqSOA both from dark-phase chemistry and OH reactions are characterized by optical properties typical of brown carbon (Laskin et al., 2015). Do the authors think that optical properties can offer further insights into cloud processing? Even though the chemistry model employed might not be able to simulate optical properties, it would be advisable to mention it, at least in the introduction, as a potentially additional tracer for aqueous phase processing.

Specific comments: Page 3: Do the authors think that back-trajectory analysis could complement the use of specific molecular tracers for the identification of specific air mass types? In addition, the backtrajetcory analysis could give an estimate of the age

of polluted air masses, to investigate the impact of fresh and aged emissions on aqSOA formation and their properties.

Table 1 could report the variability range of measurements to give an idea about the significance of differences among air mass types.

Page 10 line 337: The authors report that the size resolved composition measurements from the field experiments were noisy. Please specify if ere hygroscopicity and O/C ratios used as model input were assumed to be constant across the different size bins for the different air mass categories. In such a case, what can be the uncertainty associated with this assumption?

Figure 2: The relative mass increase calculated through equation 2 is derived for each single size bins? In such a case it is not clear why for a few size bins the dM/dlogD values are smaller than the unprocessed values, even if the relative percentage increase is significant (larger than 50

Technical comments:

Page 4 line 118: "4" in SEAC[4]RS as apice Page 8 line 250: did the authors mean "the sharp peak when SO2 is added"? Page 8 line 276: remove B after period. Page 9 line 300: stopped Page 10 line 339: Figure 2 instead of figure 1 Page 11 line 381: Figure 2 instead of figure 32 Page 15 line 509: in addition to oxalate, authors could mention also hydroxyl methane sulfonate as a tracer of aqSOA with additional chemical sinks, like oxidation under high O3 concentration (Whiteaker and Prather 2003). These sinks set some limitation on its use as a proxy for aqueous phase processing.

References: Jimenez et al., Evolution of Organic Aerosols in the Atmosphere, Science 326, 1525 (2009). Laskin et al., Chemistry of Atmospheric Brown Carbon, Chemical Reviews, 115, 4335 (2015). Whiteaker and Prather, Hydroxymethanesulfonate as a tracer for fog processing of individual aerosol particles, Atmospheric Environment, 37, 1033 (2003).

---

## Referee Comment (RC2) · Anonymous Referee #2 · 2 Jul 2018

**Review of "Is there an aerosol signature of cloud processing? " by Ervens et al. (2018)**

In the submitted manuscript, the authors present an analysis of selected field experiments and a model study based on data of the SEAC[4]RS field campaign to explore whether there is a clear aerosol signature of cloud chemistry processing. The analysis focuses on trends of changes in mass, hygroscopicity parameter $\kappa$, and oxygen-to-carbon (O/C) ratio due to chemical cloud processing.

In my opinion, the paper under discussion is often well structured, however, partly a bit lengthy. It contains some interesting information, which can provide a basis for future works on this important topic. The topic will be of interest to the journal's readers and might support future studies examining the potential role of chemical cloud processing and its impact on the CCN processing.

However, the paper in its present forms need major revision. After addressing my comments/questions/suggestions given below, this paper might be suitable for publication in ACP.

**General comments:**

Although there are interesting aspects, the whole purpose of the paper and what the authors wants to tell us, does not open to me completely. The main question of the paper, if there is an aerosol signature of chemical cloud processing, is in the end not really answered. So, does this not help anyone? The authors should discuss in the revised manuscript if at all a clear and universal answer to such a general question can be given due to the huge dependence on the specific parameters and conditions.

In the modelling section, I don't understand why there are huge differences in the input aerosol masses between the present study and the data given in Shingler et al. (2015). Moreover, the concentrations of the applied scenarios need to be discussed. Are the applied values realistic? For example, 1.25 µg m$^{-3}$ as an initial mass in the urban case seems to me very low.

The authors have introduced a new mass ratio $R_{tot}$ in order to predict the potential to extent to which aerosol properties are modified by chemical processes in clouds. However, the simple ratio is mainly dominated by the contribution of the expected $SO_2$ to sulfate formation ($R_{SO4}$) and is limited to the present aqSOA formation knowledge (individual mass yields, etc.) as well as the incomplete characterization of the OVOCs (aqSOA precursors). Therefore, to my point of view, the simple ratio does not provide much more information and does not represent a breakthrough and needed parameter.

Furthermore, the authors have outlined in the paper that not only mass-based parameters have to be considered but also parameter such as $\kappa$ and O/C ratio. However, the new mass ratio $R_{tot}$ is then purely mass-related. So, is it possible to conclude from $R_{tot}$ alone whether cloud processing will be detectable, as it is proposed in the abstract and elsewhere?

To my point of view, the conclusion section is lengthy and too little structured. I would suggest to restructure and condense this section to better present the main issues.

**Further Comments/Questions/Suggestions:**

Page1 line 1: Is the title fitting to the topic addressed in the manuscript? "cloud processing" includes also microphysical cloud processes! Therefore, please replace "cloud processing" by "chemical cloud processing" to be more precise.

Page1 line 10: Replace "Colorado" by "CO"

Page1 line 17: Replace "aqueous phase" by "aqueous-phase"

Page1 line 28: It should be noted in the abstract that the calculated $R_{tot}$ is almost exclusively dominated by the contribution of the expected $SO_2$ to sulfate formation ($R_{SO4}$). See the Table below. This finding should be also discussed in more detail in the paper.

|        | Marine | Urban | Biomass Burning | Agric. Biomass Burning | Background | Biogenic |
|--------|--------|-------|-----------------|------------------------|------------|----------|
| RSO4   | 5.1    | 1.8   | 0.1             | 0.5                    | 0.7        | 7        |
| RaqSOA | 0.005  | 0.15  | 0.03            | 0.08                   | 0.05       | 0.16     |
| total  | 5.105  | 1.95  | 0.13            | 0.58                   | 0.75       | 7.16     |
| %RSO4  | 100%   | 92%   | 77%             | 86%                    | 93%        | 98%      |
| %aqSOA | 0%     | 8%    | 23%             | 14%                    | 7%         | 2%       |

Page1 line 29: The authors have mentioned in the abstract that already "tracer compounds give evidence that aqueous phase processing occurred, they do not reveal the extent to which particle properties have been modified in terms of mass, chemical composition, hygroscopicity and oxidation state." Please, outline if the calculated parameter $R_{tot}$ leads to an improvement with regard to the prediction of a chemical cloud processing signature and help to predict the extent to which particle properties can modified in terms of mass, chemical composition, hygroscopicity and oxidation state. Please explain in detail why such a ratio, which basically only relates the gaseous $SO_2$ budget with the already existing particle mass, represents a breakthrough and needed parameter.

Furthermore, the authors should discuss in the paper that also a small mass production (low $R_{tot}$ values), due to cloud chemistry, can be important and significantly influence the aerosol properties (e.g., CNN, hygroscopic and radiative properties). For example, when surfaces of mineral dust or BC particles are modified even by a small mass addition, this can lead to significant changes in their properties. In such cases, the parameter $R_{tot}$ is definitely not the right means.

Page2 line 40: I can somehow understand the limitation of "hill cloud" experiments in terms of their geographical coverage, however, the limitations with regards to complicated interpretation due to variable advection of various sources and airmasses needs to be explained in more detail. Why should various sources and airmasses restrict the interpretation of field experiments? Maybe other limitations of hill cloud experiments can be addressed here.

Page2 line 47: Please begin the sentence with "However, SOA formation in clouds….".

Page2 line 52: Replace "aqueous phase" by "aqueous-phase"

Page2 line 57: Replace "aqueous phase" by "aqueous-phase" and "gas phase" by "gas-phase"

Page3 line 76: Replace "signature of cloud processing" by "signature of chemical cloud processing"

Page3 line 83: Replace "possible cloud processing signature" by "possible chemical cloud processing signature "

Page3 line 97: Please specify the term "monoterpenes" and list the single compounds considered here.

Page3 line 99: Here or somewhere in the text it should be mentioned that such a definition of a marine regime might include also ship emissions and urban influence. Please discuss the chosen value of 40 km. In the paper of Kummu et al. (2016), coastal continental zones are defined to be <100 km from the coast. So, I would suggest that the continental influence on the marine regime would be at least in the same range. Furthermore, the flight paths shown in the paper of Toon et al. (2016) shows just a few flights over the Gulf of Mexico, which is most likely a region with a lot of anthropogenic influence (incl. marine traffic, etc.). Furthermore, the initialized $SO_2$ concentration of 0.42 ppb is also quite high for a marine environment suggesting an anthropogenically influenced air mass. This issue should be mentioned in the manuscript. The obtained data over the ocean are maybe not representative for a pristine open ocean (marine environment). The term "Marine" is maybe not fitting here, but, has been taken over from a former study (Shingler et al., 2015).

Page4 line 114: Replace "Table S1 also lists …" by "Table S1 also lists the concentration of …"

Page4 line 140: Please use a uniform nomenclature; "Mt. Lemmon" or "Mount Lemmon", Mt. Tai etc.

Page5 line 144 and 145: Replace "gas phase" by "gas-phase"

Page5 line 151: Replace "aqueous phase" by "in-cloud" if processes in clouds are meant here.

Page5 line 159: The hygroscopic parameter needs to be introduced with a calculation formula and the corresponding reference.

Petters, M. D. and Kreidenweis, S. M.: A single parameter representation of hygroscopic growth and cloud condensation nucleus activity, Atmos. Chem. Phys., 7, 1961–1971, doi:10.5194/acp-7- 1961-2007, 2007.

Page5 line 170/171: Please do not separate the value "1" and the unit "s"

Page5 line 172: In cloud water solutions, the glyoxal and glyoxylic acid should be predominantly present in their hydrated form (gem-diol form). Thus, their O/C ratio should be 2.

Page6 line 178/179: I have compared the aerosol mass concentration given in Table 1 with the values in the cited paper of Shingler et al. (2015). There is a huge difference in the values. For example, the average total aerosol mass of the agric. biomass burning aerosol is 116.1 µg m$^{-3}$ (see Shingler et al., 2015; Figure 1) and 12.1 µg m$^{-3}$ in the present work. Please explain this difference because the $m_0$ value is an important parameter in the paper and the linked ratio $R_{tot}$. If the $m_0$ values would be larger, the calculated relative mass additions due to cloud processing would be significantly lower and, thus, the cloud signature less significant than proposed by the present model runs.

| | Marine | Urban | Biomass Burning | Agric. Biomass Burning | Background | Biogenic |
|---|---|---|---|---|---|---|
| Shingler et al. (2015) | 1.7 | 11.9 | 34.8 | 116.1 | 12.7 | 11.2 |
| Present study | 0.33 | 1.25 | 10.5 | 12.1 | 3.86 | 1.74 |
| **ratio** | **5.2** | **9.5** | **3.3** | **9.6** | **3.3** | **6.4** |

Furthermore, please explain why the total aerosol mass in the urban air mass is only 1.25 µg m$^{-3}$ and about 3.84 µg m$^{-3}$ in the background air mass. I would expect firstly much higher aerosol loadings in both cases and, secondly, lower concentrations in the less polluted background case. Thus, the concentrations of the applied scenarios need to be discussed (Are the values realistic and representative?).

Page6 line 179: Replace "gas phase" by "gas-phase"

Page6 line 179: Please correct "aerosl"

Page6 line 190/191: Here, it should also be mentioned that important sinks of organic acids may be also missing in your model, e.g., the photolysis of metal-carboxylate complexes.

Page6 line 194: Replace "aqueous phase" by "aqueous-phase"

Page6 line 199: "over the course of the 1-hr cloud simulations" should be replaced by "over the course of the 1-hr simulations" because the in-cloud time is only 40 minutes.

Page6 line 203: Replace "and thus the" by "and, thus, the"

Page6 line 204: Is the conclusion "These different time scales are in agreement with previous findings…." trivial since the same chemical mechanism has been applied in the present study?

Page6 line 205: Replace "SO$_2$ depletion" with "SO$_2$ oxidation"

Page6 line 205-207: Are the presented sulfate formation rates of ~10$^{-8}$ – 10$^{-5}$ M s$^{-1}$ an average of the first cloud period? Furthermore, a comparison with a single study is not convincing. What is the predicted pH in the different model runs (result should be provided in the SI) and what are the main S6 formation pathways in the different regimes? Are they comparable with key oxidation pathways at Mt. Tai?

Page7 line 219: Replace "aqueous phase" by "aqueous-phase"

Page7 line 227: I don't understand the following sentence, please rephrase: "AqSOA tracer compounds such as oxalate, and its main aqueous precursor glyoxylate, are clearly dominant in clouds whereas in the free troposphere organic acids dominate that significantly originate from clouds".

Page7 line 233: Replace "aqueous phase" by "aqueous-phase"

Page7 line 237: Please compare the calculated changes in κ with observed changes in the field in this subsection (see e.g., Henning et al. 2014).

Henning, S., Dieckmann, K., Ignatius, K., Schäfer, M., Zedler, P., Harris, E., Sinha, B., van Pinxteren, D., Mertes, S., Birmili, W., Merkel, M., Wu, Z., Wiedensohler, A., Wex, H., Herrmann, H., and Stratmann, F.: Influence of cloud processing on CCN activation behaviour in the Thuringian Forest, Germany during HCCT-2010, Atmos. Chem. Phys., 14, 7859-7868, https://doi.org/10.5194/acp-14-7859-2014, 2014.

Page8 line 273: Please put a space between "(2015)" and "might".

Page8 line 276: Remove "B".

Page9 line 283-297:

In general, the R$_{tot}$ ratio itself is a nice idea, but particularly the R$_{aqSOA}$ parameter is somehow quite arbitrarily defined. Furthermore, only 5 VOC precursors are taken into account in the present study.

Direct precursors for in-cloud chemical processing leading to aqSOA should be OVOCs which are only indirectly considered via their emitted precursors (such as isoprene). Furthermore, at the altitude of aircraft measurements, the emitted VOCs such as isoprene are maybe already largely oxidized to their oxidation products such as glyoxal, glycolaldehyde, MVK etc. In this case, the proposed method would require measurements of several VOCs and OVOCs. This issue should be discussed in detail. Moreover, different yields for different precursors should be used instead of a single effective mass yield factor Y which is based on one single model study focusing on isoprene. Are there other studies available which should be mentioned here?

The method applies a sum of all listed VOCs including emitted VOCs such as isoprene and important oxidation products (OVOCs) such as methyl vinyl ketone/methacrolein. If the effective mass yield factor Y is valid for isoprene, is the consideration of its oxidation products adequate?

Furthermore, it would be suitable to mention that there are also other potential precursors, which were not considered in the present study due to lacking measurements, that could contribute to aqSOA. I guess, for example, phenolic compounds can be strongly emitted by biomass burning and can contribute to aqSOA. In the marine case, the oxidation of DMS into methan sulfonic acid might be an important precursor of aqSOA. However, they are not considered in the present study. Thus, this limitation needs to be clearly addressed in the manuscript.

Overall, the assumed mass yield factor Y of 10% is of course very uncertain and the VOC/OVOC sum quite incomplete. Therefore, the authors should perform a small sensitivity study focusing on different Y values and VOC/OVOC sums to reveal the potential impact of these parameters.

Page9 line 299: Correct the values of $R_{tot}$ "0.2.0"

Page9 line 299: "Rtot" has to be subscript: "$R_{tot}$"

Page9 line 307-311: Please discuss the contribution of $R_{SO4}$ and $R_{aqSOA}$ to $R_{tot}$ in more detail and provide also some numbers in the text. Additionally, the fractions such be considered in Table 3 as shown below.

| | Marine | Urban | Biomass Burning | Agric. Biomass Burning | Background | Biogenic |
|---|---|---|---|---|---|---|
| RSO4 | 5.1 | 1.8 | 0.1 | 0.5 | 0.7 | 7 |
| RaqSOA | 0.005 | 0.15 | 0.03 | 0.08 | 0.05 | 0.16 |
| total | 5.105 | 1.95 | 0.13 | 0.58 | 0.75 | 7.16 |
| %RSO4 | 100% | 92% | 77% | 86% | 93% | 98% |
| %aqSOA | 0% | 8% | 23% | 14% | 7% | 2% |

Page9 line 310/ and Page10 line 314/342 and Page11 line 360: "R" should be "$R_{tot}$". Please check carefully the whole manuscript for missing indices.

Page11 line 360: Replace "aqueous phase" by "aqueous-phase"

Page11 line 364-379: I can somehow understand that the authors have included mainly studies from the US, however, there are also plenty of non-US studies focusing on the aerosol-cloud processing which needs to be considered here.

Page11 line 381: Please revise "Figure 32"

Page12 line 386: The abbreviation of the growth factor should be already introduced earlier in the paper (maybe in line 138). Furthermore, in the caption of Figure 6, a different abbreviation is used ("GF"). This needs to be changed or indicate the difference.

Page13 line 431/432: Do not separate "-" and "3"

Page14 line 456/457: Please cite also some experimental studies on this topic such as:

Zuo, Y. and Holgne, J.: Formation of hydrogen peroxide and depletion of oxalic acid in atmospheric water by photolysis of iron(III)-oxalato complexes, Environ. Sci. Technol., 26, 1014–1022, 1992.

Weller, C., Horn, S., Herrmann, H.: Effects of Fe(III)-concentration, speciation, excitation-wavelength and light intensity on the quantum yield of iron(III)-oxalato complex photolysis. J. Photoch. Photobio. A, 255, 41-49, 2013.

Page14 line 458: "In addition, oxalic acid and other (weaker) organic acids might evaporate from acidic aerosols." Please provide some references here.

Page14 line 470: Replace "aqueous phase" by "aqueous-phase"

Page14 line 470: "Not only aqueous phase processing in clouds but also in deliquesced aerosol particles can lead to aerosol mass." Please add some references here to performed model studies.

Page14 line 480: Replace "aqueous phase" by "aqueous-phase"

Page15 line 504: Replace "aqueous phase" by "aqueous-phase"

Page15 line 508: Replace "aqueous phase" by "aqueous-phase"

Page15 line 511: Replace "aqueous phase" by "aqueous-phase"

Page15 line 512: Replace "aqueous phase" by "aqueous-phase"

Page15 line 514: Replace "and thus" by "and, thus,"

Page15 line 519-520: Replace "signature of cloud processing" by "signature of chemical cloud processing"

Page15 line 519-520: "Overall, it can be stated that there is no unambiguous answer to the initial question in the title of this study as to whether there is a signature of cloud processing on aerosol." From my point of view, the most distinct signature present in the aerosol is still the "cloud mode" and the fact that typical secondary mass contributors mainly formed by in-cloud chemistry such as sulfate are enriched there. This cloud mode and the enrichment of sulfate can be seen, for example, from Figure 5. Thus, a signature of chemical cloud processing in aerosols is there, however, the extent to which aerosol properties are modified by chemical processes in clouds cannot be easily estimated. However, I would agree with the authors if the statement of the sentence is related to the aqSOA: "…that there is no unambiguous answer to the initial question in the title of this study as to whether there is a signature of chemical cloud processing of aqSOA on aerosol."

Page15 line 521: Please add also other important dependencies to the list such as the budget of oxidants, aerosol composition (important for the cloud pH, etc.), lifetime of clouds etc.

Page15 line 522: The authors conclude that the extent to which aerosol properties are modified by chemical processes in clouds can be quantified by the mass ratio $R_{tot}$. As mentioned in the comment above, the extent to which aerosol properties are modified depends also on other parameters. Furthermore, the current "$R_{tot}$" is almost exclusively related to the complete conversion of $SO_2$ to

sulfate and provide no size-resolved information of the possible aerosol modification. The yields of aqSOA are still an issue which requires further investigations and individual yields of aqSOA precursors are uncertain. Furthermore, the size-resolved modification of the aerosol, which is very important, can also not be quantified by $R_{tot}$. Therefore, I'm not convinced that $R_{tot}$ represents a substantial process or breakthrough to the proposed question.

If the main outcome of $R_{tot}$ is to provide a rough estimate on the question whether a mass signature of chemical cloud processing can be expected at a certain location, I'm not sure that this simple ratio will be a beneficial support for future field and model studies.

Page16 line 524: Replace "aqueous phase" by "aqueous-phase"

Page16 Reference section:

- Between the issue number of the journal and the page number is a space missing in all references. For example: " 10, 13,5839-5858, 10.5194/acp-10-5839-2010, 2010". Please correct this issue in all references.

- The doi number is missing for Aiken et al. (2008).

- The format of the doi numbers in citations is not consistent. Please revise the doi format of the references that they are consistent with the ACP format:

Please see:

https://www.atmospheric-chemistry-and-physics.net/Copernicus_Publications_Reference_Types.pdf

Felder, M., Poli, P., and Joiner, J.: Errors induced by ozone field horizontal inhomogeneities into simulated nadir-viewing orbital backscatter UV measurements, J. Geophys. Res., 112, D01303, doi:10.1029/2005JD006769, 2007.

-Page22 line 854: Please cite the final revised paper published in ACP and not the ACPD manuscript of Wonaschütz et al. 2013.

-Page21 line 838: Please correct "SEAC4RS:"

Caption of Table1: Replace "aqueous phase" by "aqueous-phase"

Caption of Table2: Please revise the line break between "(Sorooshian et al., 2007b)" and "."

Figure 1: The axis label of the hygroscopic factor $\kappa$ is covered in some case on the right y-axis (e.g. in Fig.1b).

Caption of Table 3: "(Eq-1)" should be replaced by (Equation 1) to be consistent.

Caption of Figure 2: Put a dot at the end.

Caption of Figure 4: Put a dot at the end.

Caption of Figure 5: Please use "$\kappa$" in the caption and as axis label of the y-axis.

Caption of Figure 6: Please replace "GF-derived" by "growth factor derived" and please use "$\kappa$" in the top x-axis label.

Supplemental Information:

- Add a parenthesis "log ([absolute mass increase / g m$^{-3}$])".

Caption of Table S1: Put a dot at the end.

Caption of Figure S1: Put a dot at the end. Figure S1 contains plots with the predicted relative mass concentration (dm/dlogD) and number concentration (N). Accordingly, please revise the caption or the plot.

Caption of Figure S2: The Figure contains plots of the predicted relative mass concentration (dm/dlogD) and number concentration (N). Accordingly, please revise the caption or the plot.

---

## Author Comment (AC1) · 25 Sep 2018

**Reviewer #1**

The manuscript investigates the impact of sulfate and aqueous secondary organic aerosol (aqSOA) formation through cloud processing on relative aerosol mass in- crease, aerosol hygroscopicity, and organic aerosol oxygen content, focusing first on aerosol bulk properties and then on size segregated properties. Different air mass categories are studied, using measurements collected during the SEAC4RS experiments on board the NASA DC-8, based out of-Houston. Simulations indicated that the impacts of cloud processing are more prominent on polluted air masses than clean background air masses, but less evident in heavily polluted conditions, such as in biomass burning plumes.

One of the main implications of this work is that changes in aerosol mass and particle hygroscopicity are better indicator of cloud processing than chemical markers, due to their chemistry sinks. Nevertheless, depending on the initial properties of air masses, the impact of cloud-processing on the above-mentioned parameters might vary significantly, making sometimes difficult to detect aqueous phase processing, especially under clean conditions or in heavily polluted air masses.

Cloud processing is attracting a growing interest from both the observation and modeling research communities, due to its impacts on air quality and climate. Often field observations struggle to identify aqueous phase processing, which is observed instead during laboratory experiments, delaying its description in chemistry transport models. The present manuscript helps to explain some of the discrepancies among laboratory observations, field observations, and modeling results, and deserve publication in ACP with minor revisions.

Response: We thank the reviewer for his/her constructive comments. We address them in detail in our response below. Line numbers refer to the revised manuscript without track changes.

We would like to point out that we redid some of the calculations as we noticed discrepancies between the parameters in Table 1 as compared to those reported by Shingler et al. (2016).

1) We realized that we had only listed the masses from the SMPS (D < 320 nm) in Table 1 whereas Shingler et al reported masses for the full size range (up to 850 nm). The resulting total masses are considerably higher and, thus, the resulting R values are much smaller. However, the trends for the various air masses are the same as before.

2) In the previous simulations, we had calculated the initial $\kappa$ value based on the aerosol fractions in Table 1. However, the resulting values were all significantly higher than those derived by Shingler et al. (2016) based on growth factor measurements. Several factors may have contributed to this discrepancy:

(i) The growth factors by Shingler et al., (2016) were measured for discrete sizes whereas our calculations were performed for the bulk aerosol composition. It is likely that the size-dependent values differed from the bulk values.

(ii) We assumed that all initial organics had $\kappa = 0.1$. However, fresh organics might not be hygroscopic and, thus, the kappa for the organic fraction might be smaller, in particular in fresh air masses.

(iii) Related to (ii), several studies have shown that $\kappa$ for organics at sub- and supersaturated conditions can differ significantly, with the former values being smaller. Thus, while $\kappa_{org} = 0.1$ might be appropriate close to CCN activation, the values measured at RH < 100% might be smaller.

We cannot assess which of these reasons contributed most to the discrepancy. In order to

be consistent with the measurements, we adapted the kappa values by Shingler et al. (2016) as initial bulk values.
We redid all simulations using the lower (measured) initial κ values. Accordingly, Figures 2, 4, S1 and S2 changed.

**General comments:**
The completeness of data collected during the NASA DC-8 experiments seems to be only partially deployed. For example, the HR-AMS data collected during the flights could be used to characterize the initial O/C and k parameters.
For example, the model assumes that korg is equal to 0.1, while Jimenez et al. (2009) shows that, depending on the organic oxygen content the korg can vary from about 0 up to 0.2. Please explain if the use of specific korg for different air mass types could have supported a more accurate analysis and justify why the authors decided to use a constant korg for different masses.

Response:  The reviewer is correct that a range of kappa values for organics was observed during SEAC⁴RS. However, particles with high (nearly 100%) organic content exhibited κ of ≤ ~0.1 (see Figure below). Our work is not intended to be an accurate representation of the measured values but rather seeks to show trends in expected aerosol modification. We do not expect that doing additional simulations for a range of κ will change the conclusions to a great extent.
We added the following text to the manuscript (l. 169/170):

[Figure]

*This estimate is based on measurements during SEAC⁴RS where particles with high organic content exhibited κ ~ 0.1 (Shingler et al., 2016).*

*Figure 10a by Shingler et al., 2016: (a) DASH-SP κ plotted against MF$_{OA}$ measured by the HR-AMS.*

Some authors observed that aqSOA both from dark-phase chemistry and OH reactions are characterized by optical properties typical of brown carbon (Laskin et al., 2015).
Do the authors think that optical properties can offer further insights into cloud processing? Even though the chemistry model employed might not be able to simulate optical properties, it would be advisable to mention it, at least in the introduction, as a potentially additional tracer for aqueous phase processing.
Response: We agree with the reviewer that formation of brown carbon might be used as another tracer of aqueous phase processing. However, the experiments by Laskin and others (e.g. De Haan) have been performed in solutions that are more in resemblance of aerosol particles or very concentrated cloud droplets (near formation/evaporation). Thus, such processes might occur in clouds only on very short time scales and their role for cloud processing has not been fully explored yet.
In order to account for this possibility, we added in lines 65-67 and 517, respectively:

*Several laboratory studies have shown that organics that are formed in aqueous-phase reactions might absorb light (e.g., (De Haan et al., 2010; Powelson et al., 2014; Laskin et al., 2015)). However, these products only comprise a very small fraction of the total organic carbon and are likely only formed in evaporating cloud droplets, i.e. on short time scales and when solute concentrations become sufficiently high.*

Conclusions*:… or light-absorbing products (Laskin et al., 2015)* [have been used to conclude on aqueous processing]

**Specific comments:**
**Page 3:** Do the authors think that back-trajectory analysis could complement the use of specific molecular tracers for the identification of specific air mass types? In addition, the back trajetcory analysis could give an estimate of the age of polluted air masses, to investigate the impact of fresh and aged emissions on aqSOA formation and their properties. Table 1 could report the variability range of measurements to give an idea about the significance of differences among air mass types.

Response: This is an interesting idea although we do not feel this is needed and would potentially be more distracting than helpful. We used a high volume dataset and used published criteria (Shingler et al., 2016) to categorize points into different air mass types. Conducting back-trajectory analysis on so much data would lead to a range of results and we do not view it as being helpful. However, conducting trajectory analysis is important for identifying ages of air masses and likely would be much more useful for specific case studies, but outside the scope of this study.

**Page 10 line 337:** The authors report that the size resolved composition measurements from the field experiments were noisy. Please specify if ere hygroscopicity and O/C ratios used as model input were assumed to be constant across the different size bins for the different air mass categories. In such a case, what can be the uncertainty associated with this assumption?

Response: Yes, due to the lack of additional measurements, we assumed that the initial O/C ratios and $\kappa$ values were constant throughout the size distributions. The reviewer is right that in reality there is likely some variability in hygroscopicity between the size classes but we do not think that accounting for these differences will change the main conclusions of the paper as the hygroscopicity of the CCN active particles is rather high.

**Figure 2:** The relative mass increase calculated through equation 2 is derived for each single size bins? In such a case it is not clear why for a few size bins the dM/dlogD values are smaller than the unprocessed values, even if the relative percentage increase is significant (larger than 50

Response: Activated particles increase in mass concentration due to sulfate and aqSOA addition. Thus, the mass in the initial size bins decreases while larger size bins gain in mass due to growing particles. This is the mechanism that leads to the 'Hoppel Minimum' as it was discussed, for example by (Feingold and Kreidenweis, 2000).

**Technical comments:**
**Page 4 line 118:** "4" in SEAC RS as apice
Response: We corrected it.

**Page 8 line 250:** did the authors mean "the sharp peak when SO2 is added"?

Response: We changed the text as follows (l. 257):

*The changes of $\kappa$ in the biomass burning cases (**Figure 1c** and **d**) are overall very small ($\kappa \leq 0.1$) except during the sharp peak at the beginning when sulfate is added.*

**Page 8 line 276:** remove B after period.
Response: We removed 'B'. (Note that the complete preceding sentence was removed)

**Page 9 line 300:** stopped
Response: We do not understand the reviewer's comment. No change made.

**Page 10 line 339:** Figure 2 instead of figure 1
Response: We changed the number.

**Page 11 line 381:** Figure 2 instead of figure 32
Response: We corrected the number.

**Page 15 line 509:** in addition to oxalate, authors could mention also hydroxyl methane sulfonate as a tracer of aqSOA with additional chemical sinks, like oxidation under high O3 concentration (Whiteaker and Prather 2003). These sinks set some limitation on its use as a proxy for aqueous phase processing.
Response: We agree with the reviewer that the use of HMS- as a tracer for aqueous phase processing is limited. We added the following text (l. 521):

*..or the decay or oxidation of hydroxymethanesulfonate (Kok et al., 1986; Whiteaker and Prather, 2003)*

[revised manuscript text omitted]

**Figure S 2:** Measured initial (black) and predicted cloud-processed (colored) mass distributions of aerosol particles in six air masses as identified during SEAC⁴RS. Color-coding refers to the predicted O/C ratio.

---

## Author Comment (AC2) · 25 Sep 2018

**Reviewer #2**

In the submitted manuscript, the authors present an analysis of selected field experiments and a model study based on data of the SEAC4RS field campaign to explore whether there is a clear aerosol signature of cloud chemistry processing. The analysis focuses on trends of changes in mass, hygroscopicity parameter k, and oxygen-to-carbon (O/C) ratio due to chemical cloud processing.

In my opinion, the paper under discussion is often well structured, however, partly a bit lengthy.

It contains some interesting information, which can provide a basis for future works on this important topic. The topic will be of interest to the journal's readers and might support future studies examining the potential role of chemical cloud processing and its impact on the CCN processing.

However, the paper in its present forms need major revision. After addressing my comments/questions/suggestions given below, this paper might be suitable for publication in ACP.

Response: We thank the reviewer for his/her constructive comments. We address them in detail below. Line numbers refer to the revised manuscript without track changes.

We would like to point out that we redid some of the calculations as we noticed discrepancies between the parameters in Table 1 as compared to those reported by Shingler et al. (2016).

1) We realized that we had only listed the masses from the SMPS (D < 320 nm) whereas Shingler et al reported masses for the full size range (up to 850 nm). The resulting total masses are considerably higher and, thus, the resulting R values are much smaller. However, the trends for the various air masses are the same as before.

2) In the previous simulations, we had calculated the initial $\kappa$ value based on the aerosol fractions in Table 1. However, the resulting values were all significantly higher than those derived by Shingler et al. (2016) based on growth factor measurements. Several factors may have contributed to this discrepancy:

   (i) The growth factors by Shingler et al., (2016) were measured for discrete sizes whereas our calculations were performed for the bulk aerosol composition. It is likely that the size-dependent values differed from the bulk values.

   (ii) We assumed that all initial organics had $\kappa = 0.1$. However, fresh organics might not be hygroscopic and, thus, the kappa for the organic fraction might be smaller, in particular in fresh air masses.

   (iii) Related to (ii), several studies have shown that $\kappa$ for organics at sub- and supersaturated conditions can differ significantly, with the former values being smaller. Thus, while $\kappa_{org} = 0.1$ might be appropriate close to CCN activation, the values measured at RH < 100% might be smaller.

   We cannot assess which of these reasons contributed most to the discrepancy. In order to be consistent with the measurements, we adapted the kappa values by Shingler et al. (2016) as initial bulk values.

   We redid all simulations using the lower (measured) initial $\kappa$ values. Accordingly, Figures 2, 4, S1 and S2 changed.

**General comments:**

1) Although there are interesting aspects, the whole purpose of the paper and what the authors wants to tell us, does not open to me completely. The main question of the paper, if there is an aerosol signature of chemical cloud processing, is in the end not really answered. So, does this not help anyone? The authors should discuss in the revised manuscript if at all a clear and universal answer to such a general question can be given due to the huge dependence on the specific parameters and conditions.

Response: The reviewer is right that the title question cannot be answered unambiguously. The conclusions of our study include exactly this message (l. 527)

*Overall, it can be stated that there is no unambiguous answer to the initial question in the title of this study as to whether there is always a clear signature of chemical cloud processing on aerosol. The extent to which aerosol properties are modified by chemical processes in clouds depends primarily on the initial aerosol mass, particle number concentration and sulfate and aqSOA precursor gases, as quantified by the mass ratio $R_{tot}$.*

We think that exploring our model results and comparing them to previous studies helps to identify scenarios where an aqueous phase signature can be expected. The ratio R can be used to estimate a priori whether an aqSOA, sulfate or overall cloud signature might be likely to detect when ambient experiments are planned.

2) In the modelling section, I don't understand why there are huge differences in the input aerosol masses between the present study and the data given in Shingler et al. (2015). Moreover, the concentrations of the applied scenarios need to be discussed. Are the applied values realistic? For example, 1.25 µg m-3 as an initial mass in the urban case seems to me very low.

Response: We apologize for this confusion. The masses listed in Table 1 were based on the SMSPS data only, i.e. up to ~320 nm. The model input included 'stitched' size distributions from both the SMPS and UHSAS (cf. Section 2.1.1). The full size range results in the total masses as reported by Shingler et al. (2016). The masses and resulting R values were accordingly corrected in Table 1.

3) The authors have introduced a new mass ratio Rtot in order to predict the potential to extent to which aerosol properties are modified by chemical processes in clouds. However, the simple ratio is mainly dominated by the contribution of the expected SO2 to sulfate formation (RSO4) and is limited to the present aqSOA formation knowledge (individual mass yields, etc.) as well as the incomplete characterization of the OVOCs (aqSOA precursors). Therefore, to my point of view, the simple ratio does not provide much more information and does not represent a breakthrough and needed parameter.

Response: We respectfully disagree with the reviewer that the parameter R is not a needed parameter. We are not aware that such a parameter has not been explicitly defined before in the literature. While we stated already in the conclusions that $R_{tot}$ is dominated by $R_{sulf}$, we added this information now also in the abstract (l. 31):

*As the formation processes of aqSOA are still poorly understood, the estimate of $R_{aqSOA}$ is likely associated with large uncertainties.*

Furthermore, the authors have outlined in the paper that not only mass-based parameters have to be considered but also parameter such as k and O/C ratio. However, the new mass ratio Rtot is then purely mass-related. So, is it possible to conclude from Rtot alone whether cloud processing will be detectable, as it is proposed in the abstract and elsewhere?

Response: The reviewer is right that $\kappa$ is a volume-based parameter. We think that the general trends in volume or mass increase will not differ much as they only differ by the density of the aerosol constituents. Given the differences of less than a factor of 2 in density and given all other uncertainties in the current estimates (e.g. aqSOA yield), we think that using mass- or volume-based parameters can be used interchangeably.
The O/C ratio is an atomic ratio; however, several measurements have shown that it is related to the OM/OC mass ratio and, thus, can be related to aerosol mass (e.g. Aiken et al., 2008). Thus, while not strictly a mass-based parameter, the O/C ratio can be considered being a proxy for oxidized organic aerosol mass (e.g., aqSOA).

4) To my point of view, the conclusion section is lengthy and too little structured. I would suggest to restructure and condense this section to better present the main issues.

Response: We restructured the conclusion section and shortened text where possible.

**Further Comments/Questions/Suggestions:**
**Page1 line 1:** Is the title fitting to the topic addressed in the manuscript? "cloud processing" includes also microphysical cloud processes! Therefore, please replace "cloud processing" by "chemical cloud processing" to be more precise.

Response: The reviewer is right. We added 'chemical' where appropriate.

**Page1 line 10:** Replace "Colorado" by "CO"

Response: Changed

**Page1 line 28:** It should be noted in the abstract that the calculated Rtot is almost exclusively dominated by the contribution of the expected SO2 to sulfate formation (RSO4). See the Table below. This finding should be also discussed in more detail in the paper.

| | Marine | Urban | Biomass Burning | Agric. Biomass Burning | Background | Biogenic |
|---|---|---|---|---|---|---|
| RSO4 | 5.1 | 1.8 | 0.1 | 0.5 | 0.7 | 7 |
| RaqSOA | 0.005 | 0.15 | 0.03 | 0.08 | 0.05 | 0.16 |
| total | 5.105 | 1.95 | 0.13 | 0.58 | 0.75 | 7.16 |
| %RSO4 | 100% | 92% | 77% | 86% | 93% | 98% |
| %aqSOA | 0% | 8% | 23% | 14% | 7% | 2% |

Response: We added (l. 29/30)

*$R_{tot}$ is dominated by the addition of sulfate ($R_{sulf}$) in all scenarios due to the more efficient conversion of $SO_2$ to sulfate as compared to aqSOA formation from organic gases.*

**Page1 line 29:** The authors have mentioned in the abstract that already "tracer compounds give evidence that aqueous phase processing occurred, they do not reveal the extent to which particle properties have been modified in terms of mass, chemical composition, hygroscopicity and oxidation state." Please, outline if the calculated parameter Rtot leads to an improvement with regard to the prediction of a chemical cloud processing signature and help to predict the extent to which particle properties can modified in terms of mass, chemical composition, hygroscopicity and oxidation state.
Please explain in detail why such a ratio, which basically only relates the gaseous SO2 budget with the already existing particle mass, represents a breakthrough and needed parameter.
Furthermore, the authors should discuss in the paper that also a small mass production (low Rtot values), due to cloud chemistry, can be important and significantly influence the aerosol properties (e.g., CNN, hygroscopic and radiative properties). For example, when surfaces of mineral dust or BC particles are modified even by a small mass addition, this can lead to significant changes in their properties. In such cases, the parameter Rtot is definitely not the right means.

Response: $R_{tot}$ differs in nature with the evidence provided by tracer compounds as it is a mass-based parameter rather than a qualitative marker for cloud chemistry. The reviewer is right that $R_{tot}$ (or $R_{sulf}$ or $R_{aqSOA}$, respectively) is not a direct measure of hygroscopicity. However, a high $R_{tot}$ value points to an efficient mass increase in the range of the particle distribution that has been modified by cloud processing, and thus, qualitatively to an increase of CCN sizes, hygroscopicity and change in radiative properties.

**Page2 line 40:** I can somehow understand the limitation of "hill cloud" experiments in terms of their geographical coverage, however, the limitations with regards to complicated interpretation due to variable advection of various sources and airmasses needs to be explained in more detail. Why should various sources and airmasses restrict the interpretation of field experiments? Maybe other limitations of hill cloud experiments can be addressed here.

Response: Given the rather long introduction and the fact that we refer to several key references for both airborne and hill cloud experiments, we did not add any text. The detailed description of the two types of experiments is not essential to the following discussion.

**Page2 line 47:** Please begin the sentence with "However, SOA formation in clouds….".

Response: Changed as suggested.

**Page1 line 17:** Replace "aqueous phase" by "aqueous-phase"
**Page2 line 52:** Replace "aqueous phase" by "aqueous-phase"
**Page2 line 57:** Replace "aqueous phase" by "aqueous-phase" and "gas phase" by "gas-phase"
**Page5 line 144 and 145**: Replace "gas phase" by "gas-phase"
**Page6 line 179:** Replace "gas phase" by "gas-phase"
**Page6 line 194:** Replace "aqueous phase" by "aqueous-phase"
**Page7 line 219:** Replace "aqueous phase" by "aqueous-phase"
**Page7 line 233:** Replace "aqueous phase" by "aqueous-phase"
**Page11 line 360:** Replace "aqueous phase" by "aqueous-phase"
**Page14 line 470:** Replace "aqueous phase" by "aqueous-phase"
**Page14 line 480:** Replace "aqueous phase" by "aqueous-phase"
**Page15 line 504:** Replace "aqueous phase" by "aqueous-phase"
**Page15 line 508:** Replace "aqueous phase" by "aqueous-phase"
**Page15 line 511:** Replace "aqueous phase" by "aqueous-phase"
**Page15 line 512:** Replace "aqueous phase" by "aqueous-phase"
**Page16 line 524:** Replace "aqueous phase" by "aqueous-phase"
**Caption of Table1**: Replace "aqueous phase" by "aqueous-phase"

Response: We changed 'aqueous phase' to 'aqueous-phase' and 'gas phase' to 'gas-phase', respectively, throughout the manuscript wherever it is used as an adjective.

**Page3 line 76:** Replace "signature of cloud processing" by "signature of chemical cloud processing"
**Page3 line 83:** Replace "possible cloud processing signature" by "possible chemical cloud processing signature "

Response: We added 'chemical' to cloud processing in the title and at several places throughout the manuscript

**Page3 line 97:** Please specify the term "monoterpenes" and list the single compounds considered here.

Response: The general terminology used by the PTRMS team that provides these data is to just list that measurement as "monoterpenes". The definition of this parameter in their archived read-me file is "Sum of all monoterpene isomers". Therefore, we thought it would be useful to just put in parenthesis this definition after the first time we introduce monoterpenes. Added text (l. 108):

*"…for selected species, including MACR, MVK, monoterpenes (sum of monoterpene isomers), isoprene, and acetonitrile…"*

**Page3 line 99: Here** or somewhere in the text it should be mentioned that such a definition of a

marine regime might include also ship emissions and urban influence. Please discuss the chosen value of 40 km. In the paper of Kummu et al. (2016), coastal continental zones are defined to be <100 km from the coast. So, I would suggest that the continental influence on the marine regime would be at least in the same range. Furthermore, the flight paths shown in the paper of Toon et al. (2016) shows just a few flights over the Gulf of Mexico, which is most likely a region with a lot of anthropogenic influence (incl. marine traffic, etc.). Furthermore, the initialized SO2 concentration of 0.42 ppb is also quite high for a marine environment suggesting an anthropogenically influenced air mass. This issue should be mentioned in the manuscript. The obtained data over the ocean are maybe not representative for a pristine open ocean (marine environment). The term "Marine" is maybe not fitting here, but, has been taken over from a former study (Shingler et al., 2015).

Response: The reviewer raises a good point. Many marine regions that aircraft have access to near coasts are certainly not pristine but influenced by anthropogenic pollution. For the sake of consistency with past work using the SEAC[4]RS dataset that used the same air mass categorization titles and criteria (Shingler et al., 2016; Aldhaif et al., 2018), we prefer to keep the label as Marine but to note in the text that the marine air masses sampled were not truly pristine. Added text (l. 120):

*It is further noted that the marine category is still impacted by anthropogenic pollution owing to transported continental pollution and ship exhaust and thus should not be regarded as representing pristine marine air masses.*

Response: We added the reference but did not include the equation as the parameter is commonly used in the literature.

**Page5 line 170/171:** Please do not separate the value "1" and the unit "s"

Response: The space has been removed.

**Page5 line 172:** In cloud water solutions, the glyoxal and glyoxylic acid should be predominantly present in their hydrated form (gem-diol form). Thus, their O/C ratio should be 2.

Response: The reviewer is right that in cloud water both compounds are likely hydrated. However, as organic aerosol mass, including aqSOA, is determined in dried particles, the compounds will dehydrate and thus their O/C ratio will be 1 and 1.5, respectively.

**Page6 line 178/179:** I have compared the aerosol mass concentration given in Table 1 with the values in the cited paper of Shingler et al. (2015). There is a huge difference in the values. For example, the average total aerosol mass of the agric. biomass burning aerosol is 116.1 µg m-3 (see Shingler et al., 2015; Figure 1) and 12.1 µg m-3 in the present work. Please explain this difference because the m0 value is an important parameter in the paper and the linked ratio Rtot. If the m0 values would be larger, the calculated relative mass additions due to cloud processing would be significantly lower and, thus, the cloud signature less significant than proposed by the present model runs.

| | Marine | Urban | Biomass Burning | Agric. Biomass Burning | Background | Biogenic |
|---|---|---|---|---|---|---|
| Shingler et al. (2015) | 1.7 | 11.9 | 34.8 | 116.1 | 12.7 | 11.2 |
| Present study | 0.33 | 1.25 | 10.5 | 12.1 | 3.86 | 1.74 |
| **ratio** | **5.2** | **9.5** | **3.3** | **9.6** | **3.3** | **6.4** |

Furthermore, please explain why the total aerosol mass in the urban air mass is only 1.25 µg m-3 and about 3.84 µg m-3 in the background air mass. I would expect firstly much higher aerosol loadings in both cases and, secondly, lower concentrations in the less polluted background case. Thus, the concentrations of the applied scenarios need to be discussed (Are the values realistic and representative?).

Response: Please refer to our response to the main comment 2) above. The masses in Table 1 have been corrected.

**Page6 line 179:** Please correct "aerosl"

Response: Corrected (We assume that the reviewer referred to line 186.)

**Page6 line 190/191:** Here, it should also be mentioned that important sinks of organic acids may be also missing in your model, e.g., the photolysis of metal-carboxylate complexes.

Response: We added (l. 198)

*On the other hand, loss reactions of organic acids might be underestimated, such as the photolysis of iron-dicarboxylato complexes (Weller et al., 2014).*

**Page6 line 199:** "over the course of the 1-hr cloud simulations" should be replaced by "over the course of the 1-hr simulations" because the in-cloud time is only 40 minutes.

Response: The text was changed as follows (l. 207):

*'...over the course of the 1-hr simulations, with the parcel spending about 40 min in the cloud.*

**Page6 line 203:** Replace "and thus the" by "and, thus, the"

Response: Commas were added.

**Page6 line 204:** Is the conclusion "These different time scales are in agreement with previous findings…." trivial since the same chemical mechanism has been applied in the present study?

Response: We added (l. 214)

*'...where a similar chemical mechanism was applied*

**Page6 line 205:** Replace "SO2 depletion" with "SO2 oxidation"

Response: 'Depletion' was changed 'oxidation'

**Page6 line 205-207:** Are the presented sulfate formation rates of ~10-8 – 10-5 M s-1 an average of the first cloud period? Furthermore, a comparison with a single study is not convincing. What is the predicted pH in the different model runs (result should be provided in the SI) and what are the main S6 formation pathways in the different regimes? Are they comparable with key oxidation pathways at Mt. Tai?

Response: The rates reported here are an average over the 40 min in-cloud time.
The sulfate formation pathways considered in the current study are the oxidation of S(IV) by $H_2O_2$ and $O_3$ with the former being predominant due to the moderate to low pH in the cloud water. Other

pathways have not been considered as many studies have shown that these pathways will be the dominant ones in many regions of the world. In addition, data such as metal ion concentrations (Mn, Fe) were not available from the current data set.

We would like to compare the formation rates to other studies; however, formation rates are not commonly reported.

**Page7 line 227:** I don't understand the following sentence, please rephrase: "AqSOA tracer compounds such as oxalate, and its main aqueous precursor glyoxylate, are clearly dominant in clouds whereas in the free troposphere organic acids dominate that significantly originate from clouds".

Response: We removed the sentence.

**Page7 line 237:** Please compare the calculated changes in k with observed changes in the field in this subsection (see e.g., Henning et al. 2014).
Henning, S., Dieckmann, K., Ignatius, K., Schafer, M., Zedler, P., Harris, E., Sinha, B., van Pinxteren, D., Mertes, S., Birmili, W., Merkel, M., Wu, Z., Wiedensohler, A., Wex, H., Herrmann, H., and Stratmann, F.: Influence of cloud processing on CCN activation behaviour in the Thuringian Forest, Germany during HCCT-2010, Atmos. Chem. Phys., 14, 7859-7868, https://doi.org/10.5194/acp-14-7859-2014, 2014.

Response: We thank the reviewer for this idea. We added the following discussion (l. 259ff):

*Similar trends in an increase of the hygroscopicity parameter $\kappa$ have been observed previously; for example, Henning et al. (2014) demonstrated an increase of $< 0.1 < \Delta\kappa < \sim 0.3$ in a forest site in Thuringia (Germany). However, in a marine cloud no distinct difference in hygroscopicity of particles due to cloud processing was detected (Swietlicki et al., 1999) since the pre-existing particles consisted mostly of ammonium-sulfate particles and the added mass was sulfate. In the Amazon, during the biomass burning season, the increase in particle size and hygroscopicity was small due to atmospheric processing but still more significant than in the dry season (Rissler et al., 2006). In that case, mostly organic material was added to the pre-existing particles. These trends in the various scenarios are in qualitative agreement with the findings from our model studies where the largest change in hygroscopicity is predicted to occur in biogenic areas.*

**Page8 line 273:** Please put a space between "(2015)" and "might".

Response: Space was added.

**Page8 line 276:** Remove "B".

Response: 'B' was removed. (Please note that the preceding sentence was removed as well.)

**Page9 line 283-297:**
In general, the Rtot ratio itself is a nice idea, but particularly the RaqSOA parameter is somehow quite arbitrarily defined. Furthermore, only 5 VOC precursors are taken into account in the present study.

Direct precursors for in-cloud chemical processing leading to aqSOA should be OVOCs which are only indirectly considered via their emitted precursors (such as isoprene). Furthermore, at the altitude of aircraft measurements, the emitted VOCs such as isoprene are maybe already largely oxidized to their oxidation products such as glyoxal, glycolaldehyde, MVK etc. In this case, the proposed method would require measurements of several VOCs and OVOCs. This issue should be discussed in detail. Moreover, different yields for different precursors should be used instead of a single effective mass yield factor Y which is based on one single model study focusing on isoprene. Are there other studies available which should be mentioned here?
The method applies a sum of all listed VOCs including emitted VOCs such as isoprene and important oxidation products (OVOCs) such as methyl vinyl ketone/methacrolein. If the effective mass yield factor Y is valid for isoprene, is the consideration of its oxidation products adequate? Furthermore, it would be suitable to mention that there are also other potential precursors, which were not considered in the present study due to lacking measurements, that could contribute to aqSOA. I guess, for example, phenolic compounds can be strongly emitted by biomass burning and can contribute to aqSOA. In the marine case, the oxidation of DMS into methan sulfonic acid might be an important precursor of aqSOA. However, they are not considered in the present study. Thus, this limitation needs to be clearly addressed in the manuscript.
Overall, the assumed mass yield factor Y of 10% is of course very uncertain and the VOC/OVOC sum quite incomplete. Therefore, the authors should perform a small sensitivity study focusing on different Y values and VOC/OVOC sums to reveal the potential impact of these parameters.

Response: We agree with the reviewer that the assumed yield for aqSOA is very uncertain. However, since our current study is very explorative at this point and not many lab experiments have been targeted to determine these yields, we don't think that additional model studies using different yields will lead to different conclusions. We highlighted the complexity of the aqSOA yield and the need of more experiments that refine this parameter by adding the following text (l. 308):

*This parameter might be higher for aqSOA formation from oxygenated compounds but is not a conservative value over time due to the possibly efficient decrease of aqSOA products due to oxidation. Therefore, targeted experiments should be performed in order to refine this value for a variety of (oxygenated) VOCs.*

**Page9 line 299:** Correct the values of Rtot "0.2.0"

Response: Since all R values have changed (cf introductory paragraph to this response), the value was corrected accordingly.

**Page9 line 299:** "Rtot" has to be subscript: "R$_{tot}$"

Response: Rtot was corrected to R$_{tot}$.

**Page9 line 307-311:** Please discuss the contribution of RSO4 and RaqSOA to Rtot in more detail and provide also some numbers in the text. Additionally, the fractions such be considered in Table 3 as shown below.

|             | Marine | Urban | Biomass Burning | Agric. Biomass Burning | Background | Biogenic |
|-------------|--------|-------|-----------------|------------------------|------------|----------|
| RSO4        | 5.1    | 1.8   | 0.1             | 0.5                    | 0.7        | 7        |
| RaqSOA      | 0.005  | 0.15  | 0.03            | 0.08                   | 0.05       | 0.16     |
| total       | 5.105  | 1.95  | 0.13            | 0.58                   | 0.75       | 7.16     |
| %RSO4       | 100%   | 92%   | 77%             | 86%                    | 93%        | 98%      |
| %aqSOA      | 0%     | 8%    | 23%             | 14%                    | 7%         | 2%       |

Response: We refrain from adding lines to Table 3. However, we added text to the abstract and discussion, respectively, as follows to clarify the dominant role of sulfate over aqSOA (l. 29 and 323):

*Abstract: $R_{tot}$ is dominated by the addition of sulfate ($R_{sulf}$) in all scenarios due to the more efficient conversion of $SO_2$ to sulfate as compared to aqSOA formation from organic gases.*

*Section 3.1.4: The contribution of aerosol processing by organics is highest in the biomass burning cases where $R_{aqSOA}$ is 20% and 6% of $R_{tot}$, respectively.*

**Page9 line 310/ and Page10 line 314/342 and Page11 line 360:** "R" should be "Rtot". Please check carefully the whole manuscript for missing indices.

Response: We checked the manuscript and added indices where necessary.

**Page11 line 364-379:** I can somehow understand that the authors have included mainly studies from the US, however, there are also plenty of non-US studies focusing on the aerosol-cloud processing which needs to be considered here.

Response: Most of this text refers to a study performed by one of the corresponding authors describing Figure 3 that contrasts wet vs dry conditions. In order to complement the list of other cloud processing studies, we complemented the text at the end of the paragraph (l. 392):

*Similarly, tracers of aqSOA formation were detected in fog in the Po Valley (Gilardoni et al., 2014).Cloud-processed particles were observed in many targeted field experiments. For example, increased mass in large particles was detected during the HCCT experiment where cloud processed aerosol was analysed (Henning et al., 2014).*

**Page11 line 381:** Please revise "Figure 32"

Response: Corrected.

**Page12 line 386:** The abbreviation of the growth factor should be already introduced earlier in the paper (maybe in line 138). Furthermore, in the caption of Figure 6, a different abbreviation is used ("GF"). This needs to be changed or indicate the difference.

Response: We added g(RH) when 'growth factor' is first used which is indeed at the end of section 2.1.2.

**Page13 line 431/432:** Do not separate "-" and "3"

Response: This separation cannot be changed due to the used word processor. It will be corrected in the copy-edited version of the manuscript.

**Page14 line 456/457:** Please cite also some experimental studies on this topic such as:
Zuo, Y. and Holgne, J.: Formation of hydrogen peroxide and depletion of oxalic acid in atmospheric water by photolysis of iron(III)-oxalato complexes, Environ. Sci. Technol., 26, 1014–1022, 1992.
Weller, C., Horn, S., Herrmann, H.: Effects of Fe(III)-concentration, speciation, excitation-wavelength and light intensity on the quantum yield of iron(III)-oxalato complex photolysis. J. Photoch. Photobio. A, 255, 41-49, 2013.

Response: We added these references.

**Page14 line 458**: "In addition, oxalic acid and other (weaker) organic acids might evaporate from acidic aerosols." Please provide some references here.

Response: We added the following references (Häkkinen et al., 2014; Nah et al., 2018)

**Page14 line 470:** "Not only aqueous phase processing in clouds but also in deliquesced aerosol particles can lead to aerosol mass." Please add some references here to performed model studies.

Response: We added the following references reflecting lab, field and model studies (l. 487) (Volkamer et al., 2006; Perri et al., 2010; McNeill et al., 2012; Marais et al., 2016)

**Page15 line 514:** Replace "and thus" by "and, thus,"

Response: Commas have been added.

**Page15 line 519-520:** Replace "signature of cloud processing" by "signature of chemical cloud processing"

Response: 'Chemical' was added.

**Page15 line 519-520:** "Overall, it can be stated that there is no unambiguous answer to the initial question in the title of this study as to whether there is a signature of cloud processing on aerosol." From my point of view, the most distinct signature present in the aerosol is still the "cloud mode" and the fact that typical secondary mass contributors mainly formed by in-cloud chemistry such as sulfate are enriched there. This cloud mode and the enrichment of sulfate can be seen, for example, from Figure 5. Thus, a signature of chemical cloud processing in aerosols is there, however, the extent to which aerosol properties are modified by chemical processes in clouds cannot be easily estimated.

However, I would agree with the authors if the statement of the sentence is related to the aqSOA: "…that there is no unambiguous answer to the initial question in the title of this study as to whether there is a signature of chemical cloud processing of aqSOA on aerosol."

Response: We thank the reviewer for these thoughts. We agree that the droplet mode is probably still the best indicator for cloud processing – whether by chemical or physical processes. We extended the paragraph as follows (l. 531):

*Size-resolved measurements can provide evidence whether a droplet mode exists that is formed from the addition of cloud-derived mass. Our model results show that this droplet mode can be expected to be mostly comprised of sulfate whereas a modification of aerosol properties due to aqSOA formation is likely to be small.*

**Page15 line 521:** Please add also other important dependencies to the list such as the budget of oxidants, aerosol composition (important for the cloud pH, etc.), lifetime of clouds etc.

Response: We added (l. 531):

*In addition, parameters such oxidant levels, cloud water pH and life time will also affect in-cloud mass formation rates.*

**Page15 line 522:** The authors conclude that the extent to which aerosol properties are modified by chemical processes in clouds can be quantified by the mass ratio Rtot. As mentioned in the comment above, the extent to which aerosol properties are modified depends also on other parameters.
Furthermore, the current "Rtot" is almost exclusively related to the complete conversion of SO2 to sulfate and provide no size-resolved information of the possible aerosol modification. The yields of aqSOA are still an issue which requires further investigations and individual yields of aqSOA precursors are uncertain. Furthermore, the size-resolved modification of the aerosol, which is very important, can also not be quantified by Rtot. Therefore, I'm not convinced that Rtot represents a substantial process or breakthrough to the proposed question.
If the main outcome of Rtot is to provide a rough estimate on the question whether a mass signature of chemical cloud processing can be expected at a certain location, I'm not sure that this simple ratio will be a beneficial support for future field and model studies.

Response: Given that we tried to shorten the conclusions, we did not add a further discussion of these referee comments at the end of the section. We think that $R_{tot}$ (or $R_{sulf}$ and $R_{aqSOA}$) are useful parameters to estimate the potential of air masses to introduce efficient modification of aerosol by cloud-processing. As pointed out in our responses above, Rtot does not reflect a quantitative estimate of changes in hygroscopicity or other parameters. However, it is useful as a first-order estimate for the characterization of air masses whether cloud processing can be detected or not.

**Page16 Reference section:**
- Between the issue number of the journal and the page number is a space missing in all references. For example: " 10, 13,5839-5858, 10.5194/acp-10-5839-2010, 2010". Please correct this issue in all references.

Response: We changed the references accordingly.

- The doi number is missing for Aiken et al. (2008).

Response: We added the doi number.

- The format of the doi numbers in citations is not consistent. Please revise the doi format of the references that they are consistent with the ACP format:
Please see:
https://www.atmospheric-chemistry-and-physics.net/Copernicus_Publications_Reference_Types.pdf

Felder, M., Poli, P., and Joiner, J.: Errors induced by ozone field horizontal inhomogeneities into simulated nadir-viewing orbital backscatter UV measurements, J. Geophys. Res., 112, D01303, doi:10.1029/2005JD006769, 2007.

Response: We corrected the formatting of the doi numbers accordingly

**-Page22 line 854:** Please cite the final revised paper published in ACP and not the ACPD manuscript of
Wonaschutz et al. 2013.

Response: The reference was replaced.

**-Page21 line 838:** Please correct "SEAC$^4$RS:"

Response: Corrected

**Caption of Table2:** Please revise the line break between "(Sorooshian et al., 2007b)" and "."

Response: Corrected

**Figure 1:** The axis label of the hygroscopic factor k is covered in some case on the right y-axis (e.g. in Fig.1b).

Response: We decreased the size of the legend boxes to avoid this.

**Caption of Table 3:** "(Eq-1)" should be replaced by (Equation 1) to be consistent.

Response: Eq was replaced by Equation

**Caption of Figure 2:** Put a dot at the end.
**Caption of Figure 4:** Put a dot at the end.

Response: Both captions were corrected.

**Caption of Figure 5:** Please use "k" in the caption and as axis label of the y-axis.

Response: 'kappa' was changed to 'κ'

**Caption of Figure 6:** Please replace "GF-derived" by "growth factor derived" and please use "k" in the top x-axis label.

Response: GF-derived was changed to growth-factor-derived and the figure was revised.

**Supplemental Information:**

- Add a parenthesis "log ([absolute mass increase / g m-3])".

Response: Please note that we show the absolute mass increase on a linear scale. The unit was corrected accordingly.

**Caption of Table S1:** Put a dot at the end.

Response: The period at the end was added.

**Caption of Figure S1**: Put a dot at the end. Figure S1 contains plots with the predicted relative mass concentration (dm/dlogD) and number concentration (N). Accordingly, please revise the caption or the plot.

Response: The period at the end was added.

**Caption of Figure S2:** The Figure contains plots of the predicted relative mass concentration (dm/dlogD) and number concentration (N). Accordingly, please revise the caption or the plot

Response: We changed the captions of Figure 2, S1 and S2.

**References**

[revised manuscript text omitted]

**Figure S 2:** Measured initial (black) and predicted cloud-processed (colored) mass distributions of aerosol particles in six air masses as identified during SEAC[4]RS. Color-coding refers to the predicted O/C ratio.

---

## Referee Report (RR1)

**Review of "Is there an aerosol signature of cloud processing? " by Ervens et al. (2018)**

The authors have thoroughly revised their manuscript considering mostly all of the comments raised. I have one small remaining concern about the calculated mass ratio.

I read through the manuscript, and have the following comment which need to be addressed in the final manuscript.

In the revised manuscript, the authors have used aerosols in a size range up to 850 nm for the calculation of the $R_{tot}$ factor. In the firstly submitted manuscript, aerosols in a size range up to 320 nm were used for the calculation of the $R_{tot}$ factor. The authors mentioned in their revision "The resulting total masses are considerably higher and, thus, the resulting R values are much smaller." and "Scenarios where this ratio exceeds $R_{tot}$ ~ 0.5 are the most likely ones where clouds can significantly change aerosol parameters.". The value in the firstly submitted version was $R_{tot}$ ~ 2. So, the applied aerosol size range for the calculation affects significantly the $R_{tot}$ values which are used to predict a chemical cloud-processing signature in selected air masses. Therefore, I guess it should be clearly stated in the revised manuscript that for the calculation of $R_{tot}$ values only aerosols in a size range up to 850 nm (PM 0.85) should be used. If aerosols with a different size range are used the resulting $R_{tot}$ values could be smaller or higher. Thus, a comparison with the proposed value of $R_{tot}$ (~ 0.5), provided in the present study, could be misleading.

---

## Author Response (AR2)

**Review of "Is there an aerosol signature of cloud processing? " by Ervens et al. (2018)**

The authors have thoroughly revised their manuscript considering mostly all of the comments raised. I have one small remaining concern about the calculated mass ratio.

I read through the manuscript, and have the following comment which need to be addressed in the final manuscript.

In the revised manuscript, the authors have used aerosols in a size range up to 850 nm for the calculation of the $R_{tot}$ factor. In the firstly submitted manuscript, aerosols in a size range up to 320 nm were used for the calculation of the $R_{tot}$ factor. The authors mentioned in their revision "The resulting total masses are considerably higher and, thus, the resulting R values are much smaller."

and "Scenarios where this ratio exceeds $R_{tot}$ ~ 0.5 are the most likely ones where clouds can significantly change aerosol parameters.". The value in the firstly submitted version was $R_{tot}$ ~ 2. So, the applied aerosol size range for the calculation affects significantly the $R_{tot}$ values which are used to predict a chemical cloud-processing signature in selected air masses. Therefore, I guess it should be clearly stated in the revised manuscript that for the calculation of $R_{tot}$ values only aerosols in a size range up to 850 nm (PM 0.85) should be used. If aerosols with a different size range are used the resulting $R_{tot}$ values could be smaller or higher. Thus, a comparison with the proposed value of $R_{tot}$ (~ 0.5), provided in the present study, could be misleading.

Response: We appreciate the reviewer's comment and agree with this caveat on the use of R. We ded the following text to the manuscript:

Abstract: It should be noted that the absolute value of $R_{tot}$ depends on the considered size range of particles.

At the end of Section 3.1.4: In the experiments described by Wagner et al. (2015) and Wonaschuetz et al. (2012) similar particle size ranges (< 50 nm up to 800 nm) were measured.  If a narrower range of particle sizes were taken into account (e.g., only SMPS data up to D = 316 nm, cf Section 2.1.1), the denominator in Equation-1 will be smaller and consequently the resulting R larger. Thus, by comparing R values from different experiments, it should be made sure that measurements of similar particle size ranges are considered.

In the conclusions: It should be cautioned that only measurements of similar particle size ranges should be compared since this range will determine the initial aerosol mass that is used in the calculation of R.